

# BRICK v0.1, a simple, accessible, and transparent model framework for climate and regional sea-level projections

Tony E. Wong[1,*], Alexander Bakker[1,*†], Kelsey Ruckert[1], Patrick Applegate[1,‡], Aimée Slangen[2,§], Klaus Keller[1,3,4]

[1]Earth and Environmental Systems Institute, Pennsylvania State University, University Park, PA 16802, USA
[2]Institute for Marine and Atmospheric Research Utrecht, Utrecht University, The Netherlands
[3]Department of Geosciences, Pennsylvania State University, University Park, PA 16802, USA
[4]Department of Engineering and Public Policy, Carnegie Mellon University, Pittsburgh, PA 15289, USA
* These authors contributed equally to this work.
† Now at: Rijkswaterstaat, Ministry of Infrastructure and Environment, The Netherlands
‡ Now at: Research Square, Durham, NC 27701, USA
§ Now at: Royal Netherlands Institute for Sea Research (NIOZ), Department of Estuarine & Delta Systems (EDS), Yerseke, The Netherlands
*Correspondence to*: Tony E. Wong (twong@psu.edu)

## Abstract

Simple models can play pivotal roles in the quantification and framing of uncertainties surrounding climate change and sea-level rise. They are computationally efficient, transparent, and easier to reproduce. These qualities make simple models useful for uncertainty quantification and risk characterization. Simple model codes are increasingly distributed as open source, as well as actively shared and guided. Alas, computer codes used in the geosciences can often be hard to access, run,

modify (e.g., with regards to assumptions and model components), and review. Here, we introduce a simple model framework for projections of global mean temperatures as well as regional sea levels and coastal flood risk (BRICK: Building blocks for Relevant Ice and Climate Knowledge). The BRICK model framework is written in R and Fortran and aims to help mitigate these issues, while maintaining a high degree of computational efficiency. We demonstrate the flexibility of this framework through simple model intercomparison experiments. Furthermore, we demonstrate that BRICK

is suitable for risk assessment applications by using a didactic example in local flood risk management.



# 1 Introduction

Simple, mechanistically-motivated Earth system models often play a pivotal role in climate and flood risk management (Hartin et al., 2015). For example, they are used for uncertainty quantification (Bakker et al., 2016b; Urban et al., 2014; Urban and Keller, 2010), complex model emulation (Applegate et al., 2012; Bakker et al., 2016a; Hartin et al., 2015;

Meinshausen et al., 2011a), and incorporated in integrated assessment models (Hartin et al., 2015; Meinshausen et al., 2011a).

Computational constraints often impose hard trade-offs between physical model complexity and statistical model complexity. For example, a sizable allotment of computational time could be spent running a small number of simulations

using a high-complexity physical model. Highly detailed simulations are useful to better understand the complex system, but with just a small number of simulations, only weak ensemble statistics can be drawn. In contrast, numerous realizations of a less detailed physical model could be run. This would provide the opportunity for more advanced ensemble statistical techniques including the characterization and quantification of uncertainties. It is important in climate-related applications such as mitigation of greenhouse gas emissions or adaptation to sea-level rise that the relevant uncertainties are explored and

communicated clearly to policy-makers (e.g., Garner et al., 2016; Goes et al., 2011; Hall et al., 2012; Lempert et al., 2004).

Several studies have broken important new ground in tackling these challenges. For example, Nauels et al. (2016) present a platform of sea-level emulators (i.e. simple models of complex models) that efficiently produces future projections and characterizes key model structural uncertainties using statistical calibration methods. Semi-empirical modeling (SEM)

approaches trade detailed physics for a model that can efficiently project sea level using statistical, but mechanistically motivated, relationships between sea-level changes and climate conditions such as temperature and radiative forcing (Jevrejeva et al., 2010; Rahmstorf, 2007). Recent work has expanded upon the SEM approach to use simple models to resolve individual contributions to global sea level (Bakker et al., 2016b; Mengel et al., 2016; Nauels et al., 2016).

Although there is an increasing tendency to share scientific code, it can be (perhaps surprisingly) hard to get the models running and to reproduce the results. A likely cause for this is that not enough attention is given to the scientific coding itself. Careful coding, documentation, and review require a dedicated commitment of time, but scientific incentives to do so can be weak.

Here we introduce BRICK v0.1 ("Building blocks for Relevant Ice and Climate Knowledge"), a new model framework that focuses on **accessibility, transparency, and flexibility** while maintaining, as much as possible, the computational **efficiency** that make simple models so appealing. There is a wide range of potential applications for such a model. A simple framework enables uncertainty quantification via statistical calibration approaches (Higdon et al., 2004; Kennedy and O'Hagan, 2001),



which would be infeasible with more computationally expensive models. A transparent modeling framework enables communication between scientists as well as communication with stakeholders. This leads to potential application of the model framework in decision support and education (Weaver et al., 2013). The present work expands on previous studies by (1) providing a platform of simple, but mechanistically motivated sea-level process models that resolve more processes, (2)

providing a model framework that can facilitate model comparisons (for example, between our models and those of Nauels et al. (2016)), (3) exploring combined effects of key structural and parametric uncertainties, (4) explicitly demonstrating the flexibility of our framework for interchanging model components, and (5) explicitly demonstrating the utility of our model framework for informing decision analyses.

In this model framework, we present a set of easy-to-couple simple models for climate and flood risk management. They simulate climate and contributions to global mean sea-level rise (GMSL). BRICK also includes a regional sea-level rise module, which translates the global mean sea level contributions to regional sea level at a user-defined region. We use these regional sea level projections to demonstrate how the physical model may be linked to decision-making and impacts. We implement a Bayesian calibration approach with an aim to adequately represent the tails of the distribution of future sea level

because these low-probability areas drive high-risk events. In robust decision-making approaches, it can be favorable to be underconfident as opposed to overconfident (Herman et al., 2015). We hence include a Bayesian approach with wide, mechanistically-motivated prior parameter probability distributions (Bakker et al., 2016b). Yet, its flexibility also enables the implementation of other calibration schemes. This paper is intended to showcase a useful model framework that is attractive for a sustainable approach to model development, for example by inspiring fellow researchers to contribute to the

framework, to rethink their coding practice, and maybe even to adopt some of the demonstrated design objectives in their future research proposals.

The hindcast skill of the BRICK model has been previously demonstrated (Bakker et al., 2016b). Thus, the present work focuses on outlining a set of epistemic modeling values that we believe facilitates advances in the modeling community. The

remainder of this work is organized as follows. In Sect. 2, we describe these values and the ways in which the BRICK model implementation strives to attain them. Section 3 contains an overview of the BRICK model components for climate and the contributions to sea-level rise. Section 4 describes and presents the results of a set of model experiments conducted to demonstrate how BRICK lives up to our epistemic modeling values. Section 5 summarizes the findings of this work and provides conclusions and guidance for future work.





## 2 Framework design

### 2.1 Model design

The essence of the BRICK physical model is to simulate climate change and the implied sea-level change. The socioeconomic impacts of the simulated climate and sea-level rise may then be assessed. This is depicted in Fig. 1. The

climate component, each individual contribution to global sea-level rise, and an impacts module are sub-models of BRICK, or "BRICKs." We defer details of the specific sub-models to Sect. 3. The physical model (climate and sea-level rise) components of BRICK are intentionally simple. This choice is guided by the epistemic modeling values outlined below.

### 2.2 Epistemic modeling values

#### 2.2.1 Accessibility

We selected R as the base language for BRICK because it is (1) stable, (2) freely available and open source, (3) relatively easy to use, and (4) easy to call subroutines written in faster languages. In the BRICK source code accompanying this study, the physical sub-models within the climate and sea-level rise modules are all provided as both R and Fortran 90 routines. It is our aim that the full physical-statistical model of BRICK is accessible using a modern laptop. This means that sizable Monte Carlo simulations (on the order of a million samples) must be possible on a time scale of hours. This is made possible

by calling Fortran 90 sub-models from the base code in R.

In addition to conceptual accessibility, it is our view that useful model codes are physically accessible too. Openness with scientific codes is likely to lead to higher quality codes (Easterbrook, 2014). In an effort to be truly open source and freely available, all codes – including the physical model, statistical model, and processing and plotting scripts used for the results

shown here– are available through the website URL provided in the Code Availability section of this article. Providing all code and data necessary to recreate this study is a critical component of reproducible research (Murray-Rust and Murray-Rust, 2014) and can help to build trust between the general public and scientific community (Easterbrook, 2014; Grubb and Easterbrook, 2011).

#### 2.2.2 Transparency

We aim to achieve transparency in two areas: the physical modeling, including the related model code, and the communication of scientific findings.

In regards to transparent physical modeling, we use simple numerical integration schemes whenever possible. We use as few global variables as possible, in order to "write programs for people, not computers" (Wilson et al., 2014). The essence of

these authors' advice is that users should not be expected to remember more than a few pieces of information as they read and develop code. To this end, in BRICK we aim to give appropriately suggestive names to our variables within the code,



such that a human intuitively understands what the quantity at hand represents. For example, when naming a logical or Boolean variable, we prefer for its name to read as a question that the variable itself answers, and begin the variable name with the letter "l" to imply it is a "logical" variable. One example of this in the BRICK source code is the variable "*l.project*", which is true when the model is configured to make projections of future sea-level rise and climate, and false

when the model is set up for hindcast simulations. While it may seem fussy to review these points, practices such as this will facilitate the sharing of scientific codes and enable the community to build stronger and more efficient collaborations.

Transparency also serves to link the findings of a physical model to decision-making and policy impacts. BRICK can be a useful tool to link climate changes (global temperature and sea-level rise) to decision-making frameworks through a clear

outlet for coupling to socioeconomic models. Perhaps most importantly, the coupled physical-statistical framework in BRICK incorporates many sources of uncertainty into the physical findings on which the decisions will be based. It is important that these uncertainties in climate projections are represented in the decision-making framework (Lempert et al., 2004).

### 2.2.3 Flexibility

A modular programming approach is taken with BRICK, which allows each component sub-model to be exchanged for alternative models. In this way, as the scientific forefront progresses, the BRICK sub-models may advance as well. The flexible BRICK framework also permits a quantitative evaluation of model structural differences, which is valuable in the event that it is unclear which of two candidate models should be chosen. In these cases, the BRICK framework is valuable for model comparison and quantification of structural uncertainty. As new data sets for the calibration of the sub-models

become available, these can also be incorporated instead of or in addition to the current data sets. We demonstrate the flexibility of the BRICK framework through a series of modeling experiments (Sect. 4).

### 2.2.4 Efficiency

Code efficiency is enabled primarily through (1) the use of simple models and (2) using model versions written in R for easy preliminary experimentation, and Fortran 90 versions for production simulations. This practice also follows the advice of

Wilson et al. (2014) for code developers to "write code in the highest-level language possible, and shift to lower-level languages like C and Fortran only when they are sure the performance boost is needed." This boost indeed enables the generation of production simulations on most modern laptops. The simulation of one million model iterations spanning from 1850 to present, performed on each of four CPUs (two cores and two threads per core) yields an ensemble of four million model realizations. This procedure requires less than an hour on a model year 2012 laptop with a 2.9 GHz dual-core

processor with 16 GB of RAM. Paleoclimatic simulations require longer wall clock times, but can still be completed in less than a day. All simulations for this study were completed on this machine.





Providing computationally efficient code simplifies the use. For example, there may be limitations on the computing resources allocated for a particular project, or an instructor might be interested in enhancing coursework by incorporating computer modeling exercises. In these cases, transparency is critical (as mentioned above), but also the model must be sufficiently efficient that it neither (1) expires the computational allotment for the experiment, nor (2) takes too long to be of any educational use. Our epistemic modeling values of accessibility, transparency, flexibility, and efficiency motivate the choice of a relatively simple physical modeling framework. Accordingly, a detailed statistical calibration framework is implemented. Within this framework, physical model and statistical model parameters are calibrated using observational data sets and mechanistically-motivated prior ranges. The statistical model is reviewed at greater length by Bakker et al. (2016b), so we provide only an overview in Sect. 4.1.

## 2.3 Code review and sharing

We invite the readers to download and test our code, as well as provide feedback on how best to further develop BRICK to fulfill the four epistemic values outlined above. Frequent and thorough code review by other team members as well as outside agents is another critical step towards good scientific coding practices (Wilson et al., 2014), and "peer review needs to be supplemented with a number of other mechanisms that help to establish the correctness and credibility of scientific research" (Grubb and Easterbrook, 2011). Wilson et al. (2014) also note that a number of high profile research articles have been retracted or revised because of errors in the code. The likelihood of these errors may be greatly reduced by thoroughly testing other group members' codes. In our own experience conducting the experiments described in this study, we have anecdotal evidence for the value of testing one another's code. Some errors were corrected through this process, and many more pieces of code were modified for clarity. We continue to invite all comments and suggestions for improvements and modifications (to the corresponding author).

The use of a version control system greatly expands the accessibility of a code base, and also facilitates continuous improvement of the modelling framework itself. This is true and useful before, during, and after the peer-review process. Mistakes are inevitable and we assume that BRICK still contains some minor errors, ambiguities, and pieces of code that do not fully comply to our own standards. Openly sharing the code and documentation will help to address these issues. It is our hope that BRICK may be further developed as a community modeling tool, and that other users may contribute to the framework through added or revised models and data, or improved functionality. The use of a version control system facilitates this type of community effort (Wilson et al., 2014).



## 3 Model components

### 3.1 Global mean climate

We adopt DOECLIM (Diffusion Ocean Energy balance CLIMate model, (Kriegler, 2005)) as a starting point for a simple climate model (Fig. 1). DOECLIM is an energy balance model coupled to a one-dimensional diffusive ocean model. The

DOECLIM physical model outputs are global mean surface temperature anomaly (°C) and ocean heat uptake ($10^{22}$ J). Calibration data for DOECLIM include both global surface temperature (Morice et al., 2012) and oceanic heat uptake (Gouretski and Koltermann, 2007). The required input to drive the model is the radiative forcing time series (W m$^{-2}$). This forcing is partitioned into aerosol and non-aerosol components, to enable a representation of the uncertainty associated with these forcings. The BRICK model considers this as an uncertain model parameter denoted as the aerosol forcing scaling

factor ($\alpha_{DOECLIM}$). This aerosol scaling factor has been used elsewhere in the literature (Urban et al., 2014; Urban and Keller, 2010) and accounts for some uncertainty in the radiative forcing of aerosols (Meinshausen et al., 2011b). The interested reader is directed to Kriegler (2005) and Tanaka and Kriegler (2007) for more information about the DOECLIM model.

We fit a first-order autoregressive (AR1) error model to the model-data discrepancy between temperature and ocean heat

uptake model output and calibration data. We estimate the first-order lag autocorrelation parameters ($\rho_T$ and $\rho_H$) and homoscedastic component of the AR1 innovation variance ($\sigma_T$ and $\sigma_H$) within the calibration framework as statistical model parameters. We add the heteroscedastic observational error estimates from Morice et al. (2012) and Gouretski and Koltermann (2007) in quadrature to $\sigma_T$ and $\sigma_H$ (respectively) for the complete heteroscedastic temperature and ocean heat uptake error estimates. The model calibration approach implemented here assumes normally-distributed model-data

residuals with mean zero (Higdon et al., 2004). The AR1 error model has the effect of "whitening" the residuals to satisfy this assumption.

### 3.2 Sea level components

The BRICK global mean sea-level module calculates global sea level change as the sum of four individual components: glaciers and ice caps (GIC), Greenland ice sheet (GIS), Antarctic ice sheet (AIS), and thermal expansion (TE). These

component sub-models are described in the following sections. The differential equations for the GIC, GIS, AIS, and TE contributions to global mean sea level (below) are integrated in BRICK using first-order numerical integration schemes with a one-year time step. Initial conditions are specified at a certain year. Starting from this initial condition, a first-order explicit numerical integration method integrates forward in time to the end of the simulation and a first-order implicit (backward differentiation) method integrates backward in time to the earliest year of the simulation. Preliminary experiments (not

shown) demonstrated that the one-year time step is sufficiently short to maintain numerical stability. The total global mean sea-level rise from the coupled BRICK model is



$$\frac{dS}{dt}(t) = \frac{dS_{GSIC}}{dt}(t) + \frac{dS_{GIS}}{dt}(t) + \frac{dS_{AIS}}{dt}(t) + \frac{dS_{TE}}{dt}(t),$$
(1)

where $S(t)$ is the global mean sea level (m) in year $t$, $S_{GIC}$ is the sea level contribution from GIC (m), $S_{GIS}$ is the sea level contribution from the GIS (m), $S_{AIS}$ is the sea level contribution from the AIS (m), $S_{TE}$ is the sea level contribution from thermal expansion (m). We report projections of future sea level relative to the 1986-2005 mean.

### 3.2.1 Glaciers and ice caps

We adopt a simple sub-model for the contribution to global sea-level rise from Glaciers and Ice Caps (GIC) from Wigley and Raper (2005). This same formulation is used in the MAGICC climate model (Meinshausen et al., 2011a). The parameterization for the GIC contribution to global sea-level rise is:

$$\frac{dS_{GIC}}{dt}(t) = \beta_0\big(T_g(t) - T_{eq,GIC}\big)\left(1 - \frac{S_{GIC}(t)}{V_{0,GIC}}\right)^n.$$
(2)

In Eq. (2), $S_{GIC}$ is the cumulative sea level contribution from GIC (m), $\beta_0$ is the initial mass balance sensitivity to global temperatures (m °C$^{-1}$ y$^{-1}$), $T_{eq,GIC}$ is the theoretical equilibrium temperature at which the GIC mass balance is at steady state (°C), $V_{0,GIC}$ is the initial total volume of GIC available in 1990 (m sea level equivalent (SLE)), and $n$ is an exponent parameter for area-to-volume scaling. An initial condition, $S_{0,GIC}$, is provided as an uncertain model parameter. $T_{eq,GIC}$ is taken equal to -0.15°C (Wigley and Raper, 2005). Note that in this formulation for GIC contribution to sea-level rise, whether the GIC mass is increasing or decreasing depends only on $T_g(t)$ relative to $T_{eq,GIC}$; it is independent of the current state $S_{GIC}(t)$. Within this model for the GIC sea-level contribution, $T_g$ is relative to the 1850-1870 mean global surface temperature (Wigley and Raper, 2005).

The uncertain physical model parameters for GIC-MAGICC (which will be tested in Sect. 4.2) are $\beta_0$, $V_{0,GIC}$, $S_{0,GIC}$, and $n$. We fit an AR1 model to the model-data discrepancy between GIC model output and calibration data (Dyurgerov and Meier, 2005) in the same manner as the temperature and ocean heat uptake calibration (Sect. 3.1). Uniform prior distributions are used for the GIC-MAGICC physical and statistical model parameters. These prior distributions, as well as calibrated posterior medians, 5, and 95% quantiles, are given in Appendix A.

### 3.2.2 Greenland ice sheet

BRICK uses the mechanistically-motivated SIMPLE (Simple Ice-sheet Model for Projecting Large Ensembles) model as the parameterization for the Greenland ice sheet (GIS) contribution to global mean sea level change (Bakker et al., 2016a). SIMPLE is a linear mass balance between precipitation $P_{GIS}$, runoff of meltwater $Q_{GIS}$, and the dynamic outflow of ice $D_{GIS}$,

$$\frac{dM_{GIS}}{dt} = P_{GIS} - Q_{GIS} - D_{GIS},$$
(3)



and assumes height $H_{GIS}$, volume $V_{GIS}$, mass $M_{GIS}$, and slope $sl_{GIS}$ of the ice sheet to vary proportionally. $P_{GIS}$ is often assumed to exponentially increase with temperature at mean sea-level $T_{GIS}$ by a rate of 5-7% K$^{-1}$ (e.g., Applegate et al., 2012). For a small temperature interval, this can be approximated by linearity,

$$P_{GIS}(t) = c_1 T_{GIS}(t) + c_2. \tag{4}$$

In Eqs. (4) – (7), $c_j$ are all constants, $j=1,2,...,7$. Similarly, $Q_{GIS}$ depends on the mean temperature at the ice sheet surface $T_{GIS,surface}$,

$$Q_{GIS}(t) = c_3 T_{GIS,surface}(t) + c_4, \tag{5}$$

where the difference between $T_{GIS,surface}$ and $T_{GIS}$ depends linearly on $H_{GIS}$,

$$T_{GIS,surface}(t) = T_{GIS}(t) + c_5 H_{GIS}(t). \tag{6}$$

The dynamic ice outflow is linearly dependent on the slope (thus, on the height), where the sensitivity is a function of temperature $T_{GIS}$,

$$D_{GIS}(t) = (c_6 T_{GIS} + c_7) H_{GIS} \tag{7}$$

SIMPLE (algebra) simplifies Eqs. (3) – (7) to Eqs. (8) and (9) by first estimating an equilibrium ice sheet volume ($V_{eq,GIS}$, m SLE) at which the sea level contribution from the GIS is zero, and estimating the e-folding time-scale of GIS volume changes due to changes in global temperature ($\tau_{GIS}$, y$^{-1}$).

$$V_{eq,GIS}(t) = a_{GIS} T_g(t) + b_{GIS} \tag{8}$$

$$\frac{1}{\tau_{GIS}(t)} = \alpha_{GIS} T_g(t) + \beta_{GIS} \tag{9}$$

In Eqs. (8) and (9), $a_{GIS}$, $b_{GIS}$, $\alpha_{GIS}$, and $\beta_{GIS}$ are uncertain physical model parameters. $a_{GIS}$ is the sensitivity of the equilibrium volume to changes in temperature (m SLE °C$^{-1}$); $b_{GIS}$ is the equilibrium volume $V_{eq,GIS}$ for zero temperature anomaly (m SLE); $\alpha_{GIS}$ is the sensitivity to temperature of the time-scale of GIS volume response to changes in temperature (°C$^{-1}$ y$^{-1}$);

and $\beta_{GIS}$ is the equilibrium ($T_g=0$°C) time-scale of GIS volume response to changes in temperature (y$^{-1}$). Global mean surface temperature, $T_g$, is taken relative to 1961 to 1990 mean. The GIS volume changes can then be written as

$$\frac{dV_{GIS}}{dt}(t) = \frac{1}{\tau_{GIS}(t)} \left( V_{eq,GIS}(t) - V_{GIS}(t) \right). \tag{10}$$

The initial condition $V_{0,GIS}$ is provided as an uncertain model parameter (m SLE). Using this initial condition, designated in the year 1961, the sea-level rise due to the GIS is calculated as the change from $V_{0,GIS}$ to the current GIS volume, $V_{GIS}(t)$.

This formulation, of course, assumes that all GIS volume lost makes its way into the oceans. An AR1 model is fitted to the GIS model-data residuals. Due to poor convergence, the first-order lag autocorrelation parameter ($\rho_{GIS}$) is held constant at a value determined by a preliminary model simulation that is optimized using a differential evolution optimization algorithm (Storn and Price, 1997). The GIS training data set does not provide heteroscedastic error estimates, so the AR1 innovation





variance is taken to be the estimated statistical parameter $\sigma_{GIS}$ added in quadrature to the provided error estimate (Sasgen et al., 2012). All GIS physical and statistical model parameters are assigned uniform prior distributions. The ranges for these priors and posterior distribution medians, 5, and 95% quantiles are given in Appendix A. Further details regarding SIMPLE are provided in Bakker et al. (2016a).

### 3.2.3 Antarctic ice sheet

We employ the Danish Center for Earth System Science Antarctic Ice Sheet (DAIS) model to simulate the Antarctic ice sheet contribution to global sea level (Shaffer, 2014). This is a two-dimensional model for the Antarctic ice sheet that assumes an axisymmetric geometry, shown graphically in Shaffer (2014), his Fig. 2. The DAIS model tracks changes in Antarctic ice sheet volume, considering contributions from (1) incident precipitation, (2) runoff of ice melt, (3) ice flow, and (4) ice sheet disintegration from rising and warming sea levels. Input forcings for the DAIS model include Antarctic surface temperature reduced to sea level ($T_A$, °C), high latitude ocean subsurface temperature ($T_{ANTO}$, °C), global mean sea level (m), and the time rate of change of global mean sea level (m y$^{-1}$).

When calibrated as a stand-alone model, the DAIS forcings are provided based on temperature reconstructions (see Shaffer (2014)). When the DAIS model is run as a component in the coupled BRICK model, a separate sub-model is needed to convert the global mean surface temperature from the climate model (DOECLIM) to the Antarctic surface and ocean subsurface temperatures required by the DAIS model. The Antarctic surface temperature is estimated from a linear regression with global mean surface temperature (Morice et al., 2012; Shaffer, 2014). The Antarctic ocean temperatures ($T_{ANTO}$) are modeled through a simple relation with the global mean surface temperature, $T_g$ (relative to 1850-1970 mean). $T_{ANTO}$ is bounded below at the freezing point of salt water ($T_f$= -1.4°C):

$$T_{ANTO}(t) = T_f + \frac{a_{ANTO}*T_g(t)+b_{ANTO}-T_f}{1+\exp[(a_{ANTO}*T_g(t)+b_{ANTO}-T_f)/a_{ANTO}]} \ . \tag{11}$$

Equation (11) is a modified linear regression between the global mean surface temperature $T_g$ and the Antarctic ocean temperature $T_{ANTO}$, such that the Antarctic ocean temperature is bounded below by the freezing temperature of sea water, $T_f$. In Eq. (11), $a_{ANTO}$ is the sensitivity of the Antarctic ocean temperature to global mean surface temperature (unitless), and $b_{ANTO}$ (°C) is the approximate Antarctic ocean temperature for $T_g$=0°C. $b_{anto}$ is the approximate ocean temperature because the relationship in Eq. (11) is not a simple linear regression. $a_{ANTO}$ and $b_{ANTO}$ are both estimated as uncertain model parameters. The DAIS model contains 11 physical and one statistical parameter, for a total of 14 Antarctic ice sheet parameters to be estimated.

Here, we use an updated and corrected version of the DAIS model (Ruckert et al., 2016; Shaffer, 2014). In the original formulation of the DAIS model, the input forcing from year $t$ is used to determine the AIS contribution to sea-level rise in year $t$. This implicit numerical scheme assumes sea level and temperatures for the current year are known during that model



iteration. For this study, in which temperatures and sea level originate in other BRICK model components, the DAIS model is re-cast using an explicit numerical scheme. The sea level and temperatures from the year *t-1* are used to calculate the AIS contribution in year *t*. Each mass contribution to sea level has a particular effect, or "fingerprint," on regional sea level everywhere in the ocean. We use an Antarctic shore-average fingerprint ratio of -1.0 for the AIS contribution to global sea

level, and Antarctic shore-average fingerprint factors of 1.0 for the other contributions to sea-level rise from all BRICK sub-models (Slangen et al., 2014). This Antarctic local sea level functions as the input to DAIS when run as a sub-model of the coupled BRICK model. Preliminary experiments indicated that our results are not sensitive to the precise choices of these fingerprints.

The dynamical core of the DAIS model is more detailed than the GIC, GIS, and TE emulators given above. For this reason, we do not undertake a full review of the model equations here. The interested reader is directed to Shaffer (2014) and Ruckert et al. (2016) for further details regarding the DAIS model and its hindcast forcings. Equation (3) of Shaffer (2014) is main equation of state for the Antarctic ice sheet volume ($V_{AIS}$, m³):

$$\frac{dV_{AIS}}{dt}(t) = B_{tot}(T_A, R) + F(S, R). \tag{12}$$

In Eq. (12), $B_{tot}$ is the total rate of accumulation of mass on the Antarctic ice sheet (m³ y⁻¹), $T_A$ is the Antarctic surface temperature reduced to sea level (°C), $S$ is the sea level (m), $R$ is the Antarctic ice sheet radius (m), and $F$ is the ice flux at grounding line (m³ y⁻¹). Following Shaffer (2014), we take the present sea level equivalent Antarctic ice sheet volume to be 57 m SLE, and the initial ice sheet volume ($V_{0,AIS}$, m³) to be consistent with an initial ice sheet radius of 1.86x10⁶ m. Thus, the Antarctic ice sheet contribution to global sea level may be calculated as

$$\frac{dS_{AIS}}{dt}(t) = (57\ m) * \left(1 - \frac{\frac{dV_{AIS}}{dt}(t)}{V_{0,AIS}}\right). \tag{13}$$

### 3.2.4 Thermal expansion

BRICK uses a simple parameterization for the contribution of thermal expansion (TE) of the Earth's oceans to sea-level rise. This emulator is based on the parameterizations of the sea-level rise sub-models of (Mengel et al., 2016) and was originally used by (Grinsted et al., 2010) to model the total global mean sea level changes. First, an equilibrium sea-level rise from

thermal expansion, due to changing global surface temperature ($S_{eq,TE}$, m) is calculated as

$$S_{eq,TE}(t) = a_{TE}\ T_g(t) + b_{TE}. \tag{14}$$

In Eq. (14), $a_{TE}$ is the sensitivity of the equilibrium sea-level rise from thermal expansion, due to changing global surface temperatures (m °C⁻¹), and $b_{TE}$ is the equilibrium sea-level rise from thermal expansion with no temperature anomaly (m). The sea-level rise due to thermal expansion evolves with time as

$$\frac{dS_{TE}}{dt}(t) = \frac{1}{\tau_{TE}}\left(S_{eq,TE}(t) - S_{TE}(t)\right), \tag{15}$$





where the quantity $\tau_{TE}$ is the e-folding time-scale with which current sea-level adjusts to the equilibrium state, and $1/\tau_{TE}$ is taken as an uncertain model parameter. This parameter is assigned a gamma prior distribution with shape 1.81 and scale 0.00275, which places the 5th and 95th quantiles for $\tau_{TE}$ at 82 and 1,290 years (Mengel et al., 2016). This choice of prior distribution is motivated by the fact that $\tau_{TE}$ functions similarly to the uncertain time-scale associated with an exponentially-distributed random variable. A gamma distribution is the conjugate prior for such a random variable. The initial condition $S_{0,TE}$ is provided as an uncertain model parameter (m), designated in year 1850. To match this accounting for sea-level rise relative to pre-industrial, forcing temperature is taken relative to its 1850-1870 mean. We calibrate the thermal expansion component of sea-level rise using trends reported by the International Panel on Climate Change (IPCC) Fifth Assessment Report (AR5, Church et al., 2013).

### 3.3 Regional sea-level patterns

In order to link the projections of global mean sea-level change from BRICK to a local coastal adaptation, information on regional sea level change is needed. Thus, the global mean sea level from BRICK is downscaled to regional sea level using previously published maps of scaling factors for the glacier and ice sheet components of sea-level change (Slangen et al., 2014). Any redistributions of mass between the cryosphere and the ocean (e.g. ice melt) leads not only to a change in the total mass of the ocean, but also to changes in regional sea level as a result of variations in the gravitational field of the Earth, which in turn affects the solid Earth and the rotation of the Earth (e.g., Mitrovica et al., 2001). This typically (and counterintuitively) leads to a sea-level fall close to the source of mass loss and larger-than-average sea-level rise at larger distances (> 6700 km) from the source. These so-called regional sea-level "fingerprints" are constant for the time scales used in this study, as long as the location of the ice mass change remains the same. The fingerprints can therefore be used to relate global glacier and ice sheet contributions to sea level (Sect. 3.2.1 – 3.2.3) to their regional sea level contribution.

The glacier fingerprint is based on projected changes in glacier mass in 2100 using a glacier model driven by temperature and precipitation information from the Fifth Climate Model Intercomparison Project database (Taylor et al., 2012) under the Representative Concentration Pathway 8.5 climate change scenario (RCP8.5, Moss et al., 2010), as presented in Slangen et al. (2014). It is assumed that the mass change ratios between the different glacier regions on Earth remain the same throughout the $20^{th}$ and $21^{st}$ century, which is a valid assumption as long as none of the glacier regions "finishes" (which is not expected to happen in the next century). For the Greenland and Antarctic ice sheets, it is assumed that ice melt takes place uniformly over the ice sheet surface. Within the BRICK model structure, users may define a latitude and longitude to obtain regional sea level change.





## 4 Model experiments

### 4.1 Model calibration

We calibrate the model through a coupled physical-statistical calibration framework. The relatively simple physical modeling framework of BRICK is motivated by our epistemic modeling values (Sect. 2.1). This efficient model permits the

use of a sophisticated model calibration technique. The calibration uses a robust adaptive Markov chain Monte Carlo (MCMC) approach (Vihola, 2012). The specifics of how it is applied to the BRICK model as well as a demonstration of calibrated BRICK model hindcast skill are documented in Bakker et al. (2016b).

One modification of the calibration routine relative to that presented in Bakker et al. (2016b) regards the calibration using

global mean sea level data. The vastly different time scales and characterizations of uncertainty in the Antarctic paleoclimatic calibration period and the modern period (1850 to present) lead to two separate sets of calibration parameters: (1) DAIS parameters, calibrated using paleoclimatic data, and (2) DOECLIM, GIC, GIS, and TE parameters, jointly calibrated using modern data. The paleoclimatic calibration is done using four parallel MCMC chains of 500,000 iterations each. The first 120,000 iterations of each chain are removed for burn-in. The paleoclimatic calibration requires about 10

hours on a laptop with a 2.9 GHz dual-core processor with 16 GB of RAM. The modern calibration is done using four parallel MCMC chains of $1 \times 10^6$ iterations each. The first 500,000 iterations of each chain are removed for burn-in. This requires less than one hour on the same machine as the paleoclimatic calibration. Convergence and burn-in lengths are assessed using Gelman and Rubin diagnostics (Gelman and Rubin, 1992).

We combine these two disjoint sets of parameters to form concomitant full BRICK model parameters sets, and calibrate these to global mean sea level data (Church et al., 2013) using rejection sampling (Votaw Jr. and Rafferty, 1951). In this method, each full BRICK parameter set is constructed by parsing a random draw from the calibrated DAIS parameter sets with a random draw from the DOECLIM-GIC-GIS-TE calibrated parameter sets. This full BRICK model has the major components of global mean sea-level rise represented, so only at this stage is calibration using global mean sea level data

possible. The calibration to global sea level data initially proposes 5,000 full BRICK model parameter sets. We use a joint Gaussian normal likelihood function centered at the time series of the global mean sea level data, with standard deviation given by the observational uncertainty of the sea level data. For rejection sampling, the enveloping distribution is this likelihood function evaluated at the observed sea level time series itself. Thus, no model simulation can yield a realization of the likelihood function that exceeds this value. Rejection sampling accepts each model simulation with probability equal to

the ratio of the likelihood function evaluated at the selected model simulation to the maximal value of the likelihood function. 573 ensemble members remain after the calibration to global mean sea level data. These model realizations serve as the control ensemble for analysis. The entire analysis for the control model, including paleoclimatic simulations and the risk assessment presented in Sect. 4.4 requires less than 10 minutes on a modern laptop.



In the spirit of our epistemic values, calibration routines are provided with the available BRICK source code. These routines use modern methods readily available in R. It is our aim that the interested user can easily substitute their own likelihood function (as physical scientific knowledge progresses), a new calibration method (as the statistical state-of-the-art progresses), or both. To this end, we provide a sub-routinized likelihood function, called from an R-packaged calibration method (Vihola, 2012). We also provide individual likelihood functions and calibration scripts for each sub-model of BRICK individually, to enable interested users to perform experiments using stand-alone sub-models or pre-calibration (Edwards et al., 2011).

In the interest of accessibility and transparency, with the available BRICK source code we also provide the sets of calibrated model parameters for all experiments presented here. The purpose of this is twofold. First, it greatly enhances the reproducibility of these results. Second, these data sets enable users who would like to run their own ensembles to do so. This supports our goal of accessibility. The calibrated parameter sets are provided in netCDF format, with ensemble member as the "unlimited" dimension. This permits concatenating multiple data sets by using netCDF operators (NCO) such "ncrcat" (Zender, 2008). These are freely available tools for manipulating data stored in netCDF format.

## 4.2 Exchanging BRICKs and full sea-level rise module intercomparisons

### 4.2.1 Experimental description

We achieve the accessibility, transparency, and computational efficiency of the BRICK modeling framework through use of simple models written in a simple programming environment (R). It remains to be demonstrated that this framework is flexible and efficient in post-processing.

We demonstrate BRICK's flexibility and efficiency by implementing and switching in an alternative formulation for the global mean sea level, $S(t)$. We exchange the more detailed model configuration for global mean sea level (the BRICK control, see Fig. 1) for the simple emulator described in Rahmstorf (2007). This is

$$\frac{dS}{dt}(t) = a_{GMSL}(T_g(t) - T_{eq,GMSL}) , \tag{16}$$

where $t$ is time (years), $S$ is the global mean sea level (m), $a_{GMSL}$ is a sensitivity constant (m °C$^{-1}$ y$^{-1}$), $T_g$ is the global mean surface temperature anomaly (°C), and $T_{eq,GMSL}$ is the theoretical temperature at which the global sea level is steady (°C). The parameters $a_{GMSL}$ and $T_{eq,GMSL}$, as well as the statistical parameters $\rho_{GMSL}$ (the first-order lag) and $\sigma_{GMSL}$ (the homoscedastic component of the innovation variance), are calibrated using the same global mean sea level data set as the full BRICK sea-level rise module (Church and White, 2011). The "BRICK-GMSL" model performance using Eq. (16) for the sea-level rise module (while still coupled to DOECLIM as the climate module) is compared against the full BRICK model configuration.



This BRICK-GMSL model configuration is calibrated using four parallel MCMC chains of 100,000 iterations each. The first 50,000 iterations are removed for burn-in, as determined using Gelman and Rubin diagnostics (Gelman and Rubin, 1992). We randomly sample from the resulting posterior distribution to form an ensemble for analysis of 600 model realizations. This ensemble size is chosen to be comparable with the BRICK control model ensemble size (573 members). The prior
ranges and posterior medians, 5, and 95% quantiles for the BRICK-GMSL parameters are provided in Appendix A.

Note that this specific emulator structure is arguably not the state-of-the-art anymore (Grinsted et al., 2010; Kopp et al., 2016). However, it serves here the purpose of demonstrating the ease with which alternative model formulations can be tested. This greatly simplifies, for example, model inter-comparisons and improvements. Some advantages of a simple
emulator such as this include fewer parameters to estimate and a transparent analysis. Disadvantages of such a model include the inability to resolve individual contributions to global mean sea level. This disables the use of sea level fingerprinting to obtain regional sea-level patterns. Thus, the choice of model should not only be motivated by goodness-of-fit metrics, but also by applications.

Many goodness-of-fit metrics are available for the comparison of models and data. We focus on three metrics that are motivated by the heavily-parameterized full BRICK model framework. There are 39 free parameters in the coupled climate/sea-level rise model. By contrast, BRICK-GMSL has 13 free parameters. We use the global mean sea level time series of Church and White (2011) for the model-data comparisons in skill hindcasting global mean sea level.

**Root-mean-squared-error (RMSE)** is a commonly-used error metric, so we employ it here. For consistency with other
error criteria defined below, we define the RMSE for a model as the RMSE of the model ensemble member that maximizes the likelihood function.

**Akaike Information Criterion (AIC)** is a measure of the relative goodness-of-fit between two potential models for the same data (Akaike, 1974).

$$AIC = -2\ln(L_{max}) + 2\,N \tag{17}$$

In Eq. (17), $L_{max}$ is the maximum value of the likelihood function and $N$ is the number of model parameters. Lower values of AIC provide a better match between model output and data, and consider a penalty for over-parameterizing a model.

**Bayesian Information Criterion (BIC)** is formulated similarly to AIC, but enacts a different penalty for over-parameterization (Schwarz, 1978).

$$BIC = -2\ln(L_{max}) + N\ln(N_{obs}) \tag{18}$$

In Eq. (18), $N_{obs}$ is the number of observational data points used in the model-data comparison. Thus, for $N_{obs} > e^2$, the BIC metric penalizes over-parameterization more harshly than does AIC.



### 4.2.2 Experimental results: sea-level rise module intercomparison

The full BRICK sea-level rise module (Fig. 1) performs better than the GMSL emulator (Eq. (16)) according to RMSE; the full sea-level rise module has RMSE of 0.0068 m, which is about half the GMSL emulator RMSE of 0.015 m (Fig. 2). These hindcasts are presented as sea level relative to 1961-1990 global mean sea level. This is of course expected, because the

number of free model parameters in the full BRICK model is 39, while the GMSL emulator contains only 13 free parameters. The BIC metric gives the expected result for this disparity in model complexity. The BIC for the full BRICK model with respect to the sea level data is 57.6 higher than the BIC for the GMSL emulator. The AIC is actually lower (by 17) for the full BRICK model than for the BRICK-GMSL emulator. These mixed results for the model comparison metrics indicate that using the full BRICK sea-level rise module is not unreasonably over-parameterized.

These results also show that the sea level hindcast in the full BRICK model smooths much of the year-to-year variability in sea-level rise. This can be seen by contrasting the full BRICK maximum likelihood ensemble member (solid pink line) in Fig. 2a with the BRICK-GMSL emulator maximum likelihood ensemble member in Fig. 2b. The full BRICK simulation does not capture the annual variation in global mean sea level that the BRICK-GMSL simulation successfully captures. This

does not affect ensemble statistics, however, which can be seen from the shaded envelopes around the model simulations in Fig. 2. The BRICK model has been developed with efficiency and large ensemble simulations in mind, so missing annual variability is of little concern.

This demonstrates the ease with which model intercomparisons may be undertaken using BRICK. Deactivating the glaciers

and ice caps, thermal expansion, and Greenland and Antarctic ice sheet components and integrating the GMSL emulator into BRICK involves low overhead in computer code. GMSL is the main output of the BRICK physical model. As such, it is our aim to provide a framework in which users can easily integrate new processes and models into the climate and sea-level rise modules as the scientific forefront progresses.

### 4.3 Interchanging BRICKs and sub-model intercomparisons

### 4.3.1 Experimental description

We conduct an experiment to demonstrate the flexibility of BRICK to permit easy exchanging of a single sub-model for one component of global sea-level rise. In the control BRICK model set-up, SIMPLE is used to emulate the sea-level rise contributions from the Greenland ice sheet (GIS) and GIC-MAGICC is used to emulate the contributions from glaciers and ice caps (GIC). In this model intercomparison experiment, a second version of SIMPLE is calibrated to represent the GIC

component of sea-level rise. This experiment is motivated by potential structural shortcomings of the GIC-MAGICC model. In Eq. (2), the implied GIC volume equilibrium depends only on the current surface temperature relative to the fixed parameter $T_{eq,GIC}$. If the GIC volume is quite low (almost entirely melted), this structure potentially enables unphysically fast





growth of GIC volume. The SIMPLE model (Eqs. (3) – (5)) contains an arguably more realistic representation of the relaxation of ice sheet volume towards an equilibrium. In this formulation, the time-scale of the relaxation and the equilibrium itself both depend on the surface temperature state. This type of potential disagreement within the scientific community regarding model structure is precisely where the BRICK model framework can be useful. The flexibility of

BRICK enables easy exchange of one component sub-model (GIC-MAGICC) for another (GIC-SIMPLE). This enables experiments examining the impacts of model structural choices.

This GIC-SIMPLE model configuration calibrates GIC-SIMPLE using the same observational data as the control GIC-MAGICC set-up. One key difference is that the prior distributions of the model parameters for GIC-SIMPLE were modified

to be specific to the GIC conditions instead of the GIS. These prior distributions are given in Appendix A. The same calibration method and likelihood functions are used for the GIC-SIMPLE experiment as in the GIC-MAGICC control model. We use the same basic calibration approach as in the control ensemble, which yields an ensemble of 548 model realizations for analysis in the GIC-SIMPLE experiment. As in Sect. 4.2, we focus on RMSE, AIC, and BIC as model goodness-of-fit metrics. The GIC-MAGICC model has six model parameters (four physical model, two statistical) and the

GIC-SIMPLE model has seven parameters (five physical model, two statistical).

### 4.3.2 Experimental results: glaciers and ice caps sub-model intercomparison

When the GIC-MAGICC model is used, RMSE, AIC, and BIC are all lower than when the GIC-SIMPLE model is used (Fig. 3). But the AIC and BIC are not drastically lower for GIC-MAGICC than for GIC-SIMPLE.  This indicates that the addition of a model parameter (GIC-SIMPLE) may not be justified (Kass and Raftery, 1995). The GIC contribution to global sea

level in Fig. 3 is presented relative to 1960 GIC sea-level rise. The median, 5, and 95% quantiles of the calibrated GIC-SIMPLE parameters are given in Appendix A.

The two models display similar levels of under-confidence, illustrated by the wide model ensemble envelope around the narrower range of observational data (Fig. 3) (Dyurgerov and Meier, 2005). That both models show under-confidence is

often judged to be preferable to over-confidence, especially when physical models are linked of applications-oriented decision-making frameworks (Herman et al., 2015). This experiment demonstrates BRICK's flexibility, and ability to allow the user to isolate and examine any source of uncertainty or dissatisfaction in the modeling framework. These results also provide guidance for the use of the BRICK model framework for model intercomparison and selection experiments. At present we do not make any recommendations regarding which GIC sub-model to use. The GIC-MAGICC component has

both strengths (e.g., fewer parameters and appropriate in melting regimes) and weaknesses (unphysical GIC growth, does not encourage growth beyond $V_{0,GIC}$, state-independent equilibrium).





### 4.4 Linking an impacts and decision-analysis module to BRICK

### 4.4.1 Experimental description

We demonstrate the ability of the BRICK framework to incorporate additional structure to link the physical model for surface temperature and sea-level rise (climate and sea level modules, Fig. 1) to socioeconomic implications (impacts
module, Fig. 1). In this example application, we use the calibrated ensemble in the BRICK control configuration to obtain local sea level projections for New Orleans, Louisiana. We use a common didactic model for coastal flood protection (Van Dantzig, 1956; Jonkman et al., 2009). In this flood risk model, the policy lever available to decision-makers is the amount by which to heighten the dikes protecting the coastal community. We consider a previously published simple analysis that focuses on the northern dike ring in central New Orleans (Jonkman et al., 2009). We use this illustrative cost-benefit
approach to calculate an economically-efficient dike-heightening by weighing the decrease in probable losses due to flooding achieved by building taller dikes against the increase in costs due to investments in construction.

The flood risk model implemented here follows a commonly used simple approach (Van Dantzig, 1956). The present implementation considers the current year as 2015 and a time horizon of 85 years (to 2100). We consider discrete dike
heightenings in increments of 5 centimeters, between 0 and 10 meters. The average annual flood probability is calculated from the simulated local sea-level rise, the land subsidence rate (Dixon et al., 2006), and flood frequency parameters (Jonkman et al., 2009). We calculate the expected losses (US dollars) for each proposed dike heightening from the flood probabilities for each heightening, the value of goods protected by the dike ring, and the net discount rate (Jonkman et al., 2009). We use a linear approximation of the investment to achieve a particular dike heightening from previous work to
calculate expected investments for each dike heightening (Jonkman et al., 2009). The total expected costs are the sum of the expected losses and the expected investments. With respect to dike heightening, the expected investments are a linearly increasing function and the expected losses are an exponentially decreasing function. The minimum total expected cost then is the economically-efficient dike heightening strategy in the framework of this simple illustrative model (Eq. (14) of Van Dantzig (1956)).

The uncertain parameters considered in this cost-benefit analysis include the initial flood frequency with no heightening (y$^{-1}$); the exponential flood frequency constant (m$^{-1}$); the value of goods protected by the dike ring (billion US dollars); the net discount rate (%); the uncertainty in investment costs (a unitless multiplicative factor); and the land subsidence rate (m y$^{-1}$). We sample the uncertainty in these parameters via Latin hypercube, where the population size is given by the number of sea-
level rise ensemble members that are present (573 for the control BRICK ensemble). The distributions from which the economic model parameters are drawn are given in Table 2. Each realization of regional sea level is assigned a concomitant sample of flood risk model parameters. An economically-efficient dike heightening is calculated for each ensemble member. "Return periods" (years) correspond to the frequency of storms with the potential to overtop dikes with the corresponding



dike height – essentially, the inverse of the annual flood probability. Return periods are a convenient and intuitive way to view the probabilities of flooding in this economic analysis.

We present results for the flood risk management experiment using sea level projections under the Representative Concentration Pathway (RCP) 8.5. We note that many factors are not incorporated into this analysis and this simple illustration is not designed to be used for real decision making. For example, storm surge and structural failure are not considered (Grinsted et al., 2013; Moritz et al., 2015). The purpose of this illustration is to demonstrate the flexibility and transparency of the BRICK model framework. This experiment highlights the importance of transparency in particular when linking physical modeling results to the impacts on socioeconomic modeling and policy decision-making.

### 4.4.2 Experimental results: regional sea-level changes

Regional sea level is projected to 2100 under the climate change scenarios of RCP2.6, 4.5, and 8.5 (Fig. 4). These projections use the control configuration of the model, with GIC-MAGICC and the full sea-level rise sub-model set-up depicted in Fig. 1. The ensemble median projection is shown in Fig. 4. Sea level rises by 2100 globally by about 50 cm, 71 cm, and 126 cm under RCP2.6, 4.5, and 8.5, respectively (ensemble median). The Arctic Ocean is an obvious exception to the rest of the ocean. Due to the Greenland ice mass loss, Arctic regional sea level will fall as a result of the loss of gravitational attraction. However, the addition of mass raises sea level in other parts of the ocean farther away. Arctic sea level (median sea level of all latitudes higher than 60°N) increases by 7 cm under RCP2.6, but falls by 1 cm under RCP4.5 and by 28 cm under RCP8.5. By contrast, the tropical sea level (median of all latitudes between 30°S and 30°N) rises by 55 cm, 79 cm, and 142 cm under RCP2.6, 4.5, and 8.5, respectively, which is greater than the global mean rise. Due to the asymptotically increasing gravitational effects in proximity to the melting Greenland ice sheet, sea-level fall below -1.5 m is cut off at -1.5 m.

### 4.4.3 Experimental results: Link to coastal defense strategies

Regional sea level projections scaled to New Orleans, Louisiana are used in a common flood risk management example. We find the economically-efficient (i.e., cost-minimizing) dike heightening to be 1.45 m (ensemble mean; 90% range is 0.75 to 1.95 m). This heightening corresponds to a return period of about 1270 years (ensemble mean; 90% range is roughly 200-3000 years). The simple analysis presented here should not be used to inform on-the-ground decisions in New Orleans. This experiment is meant to demonstrate BRICK's ability to contribute in risk assessment applications.

### 5 Conclusions

We present BRICK v0.1, a modeling framework for global and regional sea-level change. BRICK has been designed with four epistemic modeling goals: **accessibility**, **transparency**, **efficiency**, and **flexibility**. BRICK can skillfully match





observational data for individual sea level contributions in hindcast (Bakker et al., 2016b). Here we focus on how BRICK achieves our epistemic values using a set of modeling experiments.

BRICK is coded in the widely available and simple coding language R, to achieve the goals of accessibility and transparency. The main physics (climate and sea-level rise) codes are also (redundantly) transcribed in Fortran 90, for more efficient simulations. BRICK is designed to be transparent, as well as efficient, by coupling previously published simple, mechanistically motivated models for the major contributors to global sea level. The efficient physical modeling approach provides the opportunity to incorporate a rigorous statistical calibration framework as well, wherein various sources of uncertainty are incorporated into model projections (see Bakker et al. (2016b) for a more detailed discussion of this). Finally, the model comparison experiments in Sect. 4.2 and 4.3 demonstrate the flexibility of the BRICK modeling framework. These sections bring into focus the importance of these epistemic modeling values. A modeling framework that is (in particular) transparent and accessible can help to streamline the process of quantifying the local impacts of the physical model results, to link to decision-analytical models, and to communicate these results to stakeholders and decision-makers.

We hope that the accessibility and transparency of BRICK are helpful to others, and will stimulate the continuous peer-reviewing, challenging, and improving of the BRICK framework. Of course, although we tried to couple models that fit our epistemic model values as close as possible, we assume that others may prefer other models and may have different epistemic values. Our framework is designed in such a way that it is possible to plug in other model components to reflect these different values. For example, it would be very interesting to add the components models used for the semi-empirical model frameworks of Mengel et al. (2016) and Nauel et al. (2016).

We demonstrated the flexibility and transparency of BRICK in connecting projections from the physical model to the impacts on a local risk and decision-analysis problem. The simple probabilistic calibration method and cost-benefit analysis that we adopted for the simple demonstration can be expanded to incorporate aspects of deep uncertainties (Lempert et al., 2004; Weaver et al., 2013) as well as more complex decision-making frameworks (e.g., considering multiple objectives, beyond only expected total costs) (Kasprzyk et al., 2013; Lempert, 2014; Lempert and Collins, 2007). Climate change poses decision problems where strong connections across academic disciplines are critical. Further, the study of climate modeling relies on communal modeling efforts. The need for transparent communication among modelers and between disciplines is where the BRICK framework and the epistemic modeling values presented here can facilitate future developments. Above all, we hope that BRICK inspires the involved communities to pay careful attention to enhance flexibility, transparency, and accessibility of modelling frameworks.



**Code and Data Availability**

All BRICK0.1 code is available at http://download.scrim.psu.edu/Wong_etal_BRICK under the GNU general public open source license. The data sets as well as the calibration methods used for model comparisons and calibrations in this study are provided along with the model code.





## Appendix A: Prior probability distribution ranges for the sub-model parameters, and median, 5th, and 95th quantiles of the calibrated posterior parameter distributions.

**Table A1.** Prior probability distribution ranges for the DOECLIM climate model parameters, and median, 5th, and 95th quantiles of the calibrated posterior parameter distributions. The priors are all uniformly distributed.

| Parameter | Units | Lower bound | Upper bound | 5% | Median | 95% |
|---|---|---|---|---|---|---|
| $S$ | °C | 0.1 | 10 | 1.6 | 2.3 | 3.6 |
| $\kappa_{DOECLIM}$ | cm$^2$ s$^{-1}$ | 0.1 | 4 | 0.39 | 1.6 | 3.6 |
| $\alpha_{DOECLIM}$ | [-] | 0 | 2 | 0.44 | 0.78 | 1.1 |
| $T_0$ | °C | -0.3 | 0.3 | -0.083 | -0.038 | 0.0050 |
| $H_0$ | $10^{22}$ J | -50 | 0 | -42 | -30 | -15 |
| $\sigma_T$ | °C | 0.05 | 5 | 0.070 | 0.080 | 0.092 |
| $\sigma_H$ | $10^{22}$ J | 0.1 | 10 | 0.18 | 1.0 | 2.9 |
| $\rho_T$ | [-] | 0 | 0.999 | 0.062 | 0.5 | 0.95 |
| $\rho_H$ | [-] | 0 | 0.999 | 0.36 | 0.48 | 0.61 |

**Table A2.** Prior probability distribution ranges for the thermal expansion model parameters, and median, 5th, and 95th quantiles of the calibrated posterior parameter distributions. The prior distribution for $1/\tau_{TE}$ is a gamma distribution (see main text). The other priors are all uniformly distributed.

| Parameter | Units | Lower bound | Upper bound | 5% | Median | 95% |
|---|---|---|---|---|---|---|
| $a_{TE}$ | m °C$^{-1}$ | 0 | 0.8595 | 0.11 | 0.41 | 0.80 |
| $b_{TE}$ | m | 0 | 2.193 | 0.035 | 0.34 | 1.5 |
| $1/\tau_{TE}$ | y$^{-1}$ | 0 | 1 | 0.00046 | 0.0017 | 0.0056 |
| $S_{0,TE}$ | m | -0.0484 | 0.0484 | -0.044 | 0.0014 | 0.043 |





**Table A3.** Prior probability distribution ranges for the GIS-SIMPLE Greenland ice sheet model parameters, and median, 5th, and 95th quantiles of the calibrated posterior parameter distributions. The priors are all uniformly distributed. Due to convergence issues, $\rho_{GIS}$ is held fixed at a value calculated from a preliminary optimized model simulation (see main text).

| Parameter | Units | Lower bound | Upper bound | 5% | Median | 95% |
|---|---|---|---|---|---|---|
| $a_{GIS}$ | m °C$^{-1}$ | -4 | -0.001 | -3.9 | -3.0 | -1.6 |
| $b_{GIS}$ | m | 5.888 | 8.832 | 7.4 | 7.8 | 8.1 |
| $\alpha_{GIS}$ | °C$^{-1}$ y$^{-1}$ | 0 | 0.001 | 0.00038 | 0.00075 | 0.00098 |
| $\beta_{GIS}$ | y$^{-1}$ | 0 | 0.001 | 2.3x10$^{-5}$ | 0.00013 | 0.00041 |
| $V_{0,GIS}$ | m | 7.16 | 7.56 | 7.2 | 7.4 | 7.5 |
| $\sigma_{GIS}$ | m | 0 | 0.002 | 0.00017 | 2.0x10$^{-4}$ | 0.00025 |
| $\rho_{GIS}$ | [-] | [-] | [-] | [-] | 0.92 | [-] |

5   **Table A4.** Prior probability distribution ranges for the DAIS Antarctic ice sheet model parameters, and median, 5th, and 95th quantiles of the calibrated posterior parameter distributions. An inverse gamma prior distribution is used for $\sigma^2_{DAIS}$ (see Ruckert et al. (2016)). All other prior distributions are uniform.

| Parameter | Units | Lower bound | Upper bound | 5% | Median | 95% |
|---|---|---|---|---|---|---|
| $a_{ANTO}$ | °C °C$^{-1}$ | 0 | 1 | 0.034 | 0.40 | 0.96 |
| $b_{ANTO}$ | °C | 0 | 2 | 0.12 | 1.1 | 1.9 |
| $\gamma$ | [-] | 0.5 | 4.25 | 1.4 | 3.1 | 4.1 |
| $\alpha_{DAIS}$ | [-] | 0 | 1 | 0.040 | 0.39 | 0.78 |
| $\mu$ | m$^{1/2}$ | 7.05 | 13.65 | 7.4 | 11 | 13 |
| $\nu$ | m$^{-1/2}$ y$^{-1/2}$ | 0.003 | 0.015 | 0.0038 | 0.0089 | 0.014 |
| $P_0$ | m y$^{-1}$ | 0.026 | 1.5 | 0.15 | 0.50 | 1.2 |
| $\kappa_{DAIS}$ | °C$^{-1}$ | 0.025 | 0.085 | 0.029 | 0.057 | 0.082 |
| $f_0$ | m y$^{-1}$ | 0.6 | 1.8 | 0.7 | 1.3 | 1.8 |
| $h_0$ | m | 735.5 | 2206.5 | 1100 | 1700 | 2200 |
| $C$ | m °C$^{-1}$ | 47.5 | 142.5 | 51 | 80 | 120 |
| $b_0$ | m | 740 | 820 | 750 | 780 | 820 |
| $slope$ | [-] | 0.00045 | 0.00075 | 0.00056 | 0.00065 | 0.00074 |
| $\sigma^2_{DAIS}$ | m$^2$ SLE | 0 | [-] | 0.19 | 0.49 | 2.0 |





**Table A5.** Prior probability distribution ranges for the GIC-MAGICC glaciers and ice caps model parameters, and median, 5th, and 95th quantiles of the calibrated posterior parameter distributions. The priors are all uniformly distributed.

| Parameter | Units | Lower bound | Upper bound | 5% | Median | 95% |
|---|---|---|---|---|---|---|
| $\beta_{GIC}$ | m y$^{-1}$ °C$^{-1}$ | 0 | 0.041 | 0.00057 | 0.00089 | 0.0014 |
| $V_{0,GIC}$ | m | 0.3 | 0.5 | 0.31 | 0.40 | 0.49 |
| $N$ | [-] | 0.55 | 1 | 0.57 | 0.79 | 0.97 |
| $S_{0,GIC}$ | m | -0.0041 | 0.0041 | -0.0038 | $3.0 \times 10^{-6}$ | 0.0037 |
| $\sigma_{GIC}$ | m | 0 | 0.0015 | $1.9 \times 10^{-5}$ | 0.00021 | 0.00063 |
| $\rho_{GIC}$ | [-] | -0.999 | 0.999 | 0.30 | 0.87 | 0.99 |

**Table A6.** Prior probability distribution ranges for the GIC-SIMPLE model parameters, and median, 5th, and 95th quantiles of the calibrated posterior parameter distributions. The priors are all uniformly distributed.

| Parameter | Units | Lower bound | Upper bound | 5% | Median | 95% |
|---|---|---|---|---|---|---|
| $a_{GIC}$ | m °C$^{-1}$ | -4 | -0.001 | -3.60 | -1.90 | -0.73 |
| $b_{GIC}$ | m | 0.3 | 0.5 | 0.31 | 0.39 | 0.49 |
| $\alpha_{GIC}$ | °C$^{-1}$ y$^{-1}$ | 0 | 0.001 | $4.3 \times 10^{-5}$ | 0.00044 | 0.00093 |
| $\beta_{GIC}$ | y$^{-1}$ | 0 | 0.001 | $7.9 \times 10^{-5}$ | 0.00048 | 0.00094 |
| $V_{0,GIC}$ | m | 0.3 | 0.5 | 0.31 | 0.41 | 0.49 |
| $\sigma_{GIC}$ | m | 0 | 0.0015 | $1.4 \times 10^{-5}$ | 0.00020 | 0.00065 |
| $\rho_{GIC}$ | [-] | -0.999 | 0.999 | 0.55 | 0.90 | 0.99 |

**Table A7.** Prior probability distribution ranges for the Rahmstorf (2007) global mean sea level model parameters, and median, 5th, and 95th quantiles of the calibrated posterior parameter distributions. The priors are all uniformly distributed.

| Parameter | Units | Lower bound | Upper bound | 5% | Median | 95% |
|---|---|---|---|---|---|---|
| $a_{GMSL}$ | m °C$^{-1}$ | 0 | 0.0035 | 0.0012 | 0.0020 | 0.0030 |
| $T_{eq,GMSL}$ | m | -1.5 | 1.5 | -1.2 | -0.58 | -0.28 |
| $\sigma_{GMSL}$ | m | 0 | 0.05 | $7.0 \times 10^{-5}$ | 0.00075 | 0.0019 |
| $\rho_{GMSL}$ | [-] | 0 | 0.999 | 0.35 | 0.62 | 0.88 |



**Author Contributions**

K. Keller initiated the study. A. Bakker and T. Wong designed the general framework and research. T. Wong and A. Bakker designed the initial figures and wrote the first draft. T. Wong, A. Bakker, K. Ruckert, and P. Applegate produced the major part of the coding and code testing. A. Slangen produced and interpreted the regional sea level fingerprinting data. All contributed to the final text.

**Acknowledgments**

We gratefully acknowledge Jared Oyler for guidance during code development. We thank Rob Nicholas, Chris and Bella Forest, Nancy Tuana, Robert Lempert, and Gary Shaffer for helpful contributions. This work was partially supported by the National Science Foundation through the Network for Sustainable Climate Risk Management (SCRiM) under NSF cooperative agreement GEO-1240507 as well as the Penn State Center for Climate Risk Management. Any conclusions or recommendations expressed in this material are those of the authors and do not necessarily reflect the views of the funding agencies. Any errors and opinions are, of course, those of the authors.



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





**Tables**

**Table 1.** Parameter descriptions and prior probability distributions for flood protection cost-benefit analysis.

| Parameter | Description | Distribution |
|---|---|---|
| $p_0$ | Initial flood frequency ($yr^{-1}$) with zero heightening | log-N(log-$\mu$=log(0.0038), log-$\sigma$=0.25) |
| $\alpha$ | Exponential flood frequency constant ($m^{-1}$) | N($\mu$=2.6, $\sigma$=0.1) |
| V | Value of goods protected by dike ring (billion US$) | U(5, 30) |
| $\delta$ | Net discount rate (-) | U(0.02, 0.06) |
| $I_{unc}$ | Investment uncertainty (-) | U(0.5, 1) |
| $r_{subs}$ | Land subsidence rate (m $yr^{-1}$) | log-N(log-$\mu$=log(0.0056), log-$\sigma$=0.4) |





**Figures**

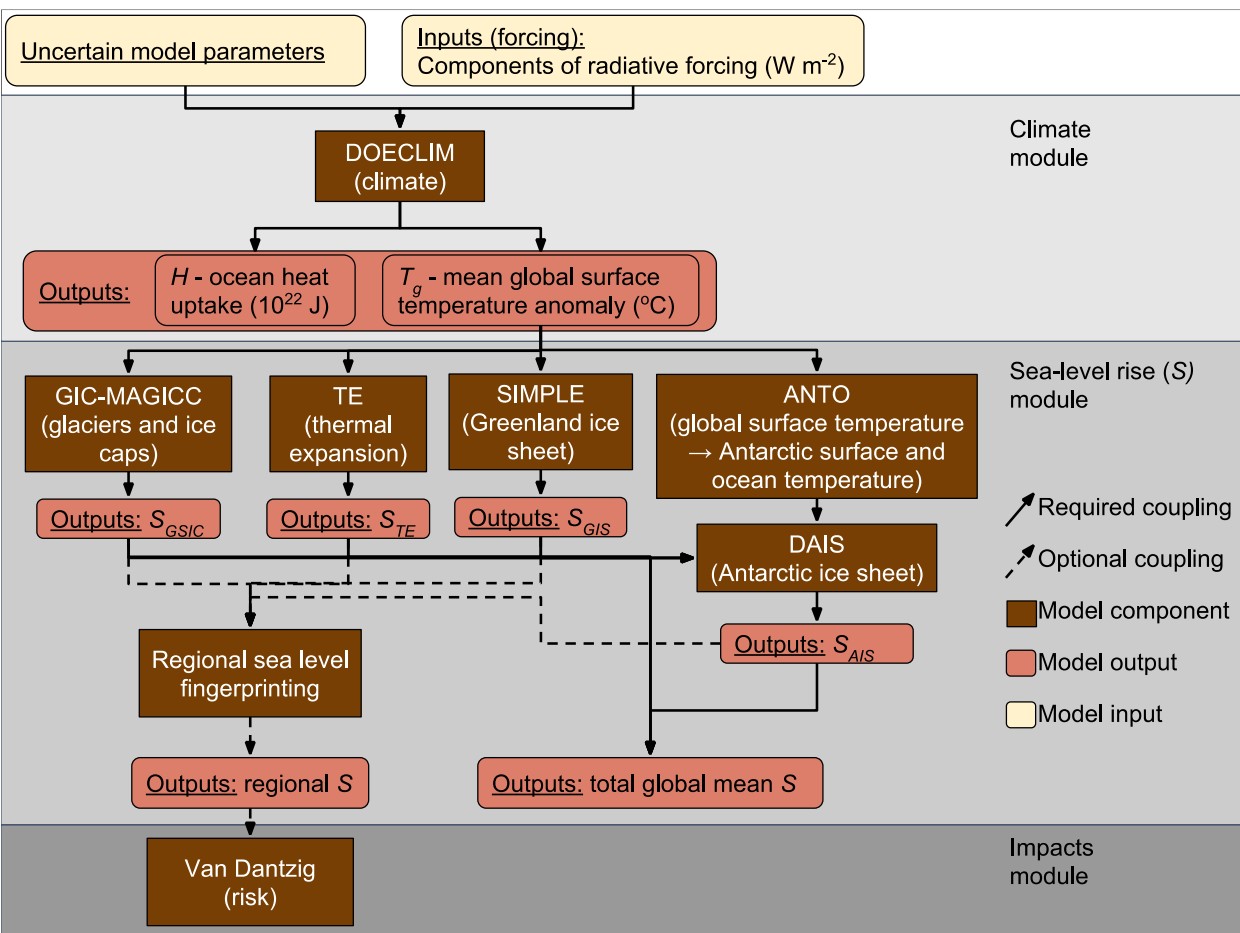

**Figure 1.** BRICK model structural diagram. Dashed connectors indicate couplings that are non-essential for projections of global mean sea level. These dashed couplings are required for projecting regional sea level and climate impacts. DOECLIM is the Diffusion-Ocean-Energy balance CLIMate model (Kriegler, 2005); GIC-MAGICC is the Glaciers and Ice Caps module from the climate model MAGICC (Meinshausen et al., 2011a); TE is the Thermal Expansion model (Grinsted et al., 2010; Mengel et al., 2016); SIMPLE is the Simple Ice-sheet Model for Projecting Large Ensembles (Bakker et al., 2016a); ANTO is the ANTarctic Ocean temperature model; DAIS is the Danish Center for Earth System Science Antarctic Ice Sheet model (Shaffer, 2014); regional sea level fingerprinting downscales from global sea-level contributions to regional (Slangen et al., 2014); and the model of Van Dantzig (1956) assesses flood risk.



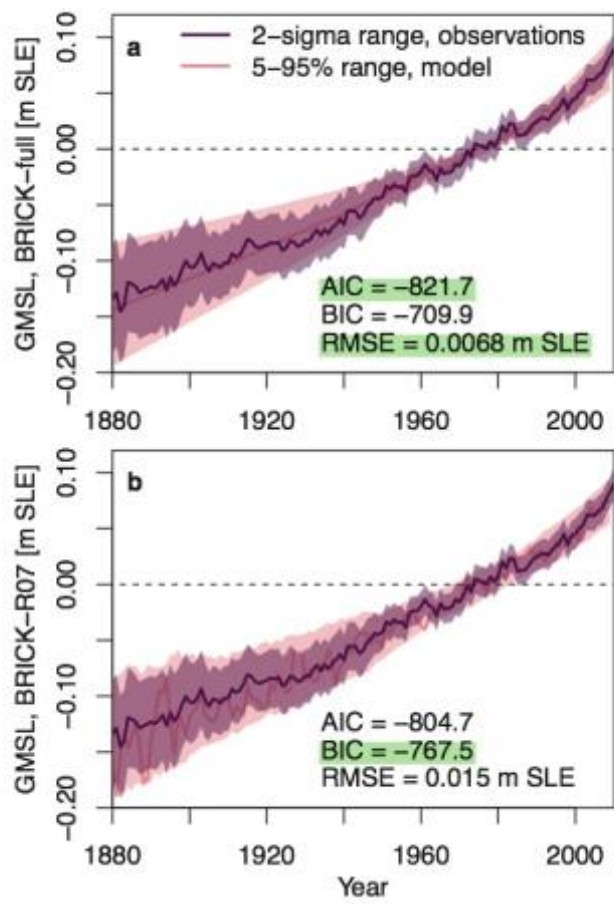

**Figure 2.** Comparison of global mean sea-level rise hindcast skill relative to sea level data (Church and White, 2011), using (a) the full sub-model approach (GIC, GIS, TE, and AIS) and (b) the model for global mean sea-level rise of Rahmstorf (2007). Sea level is relative to 1961-1990 global mean sea level. Both model configurations use DOECLIM as the climate module. Lower values of Akaike Information Criterion (AIC), Bayesian Information Criterion (BIC), and root-mean-squared error (RMSE) indicate a better model fit to the data. These error metrics are all calculated using the maximum likelihood ensemble member, which is represented by the solid pink line. Green highlighting indicates the model structure suggested by each comparison metric.



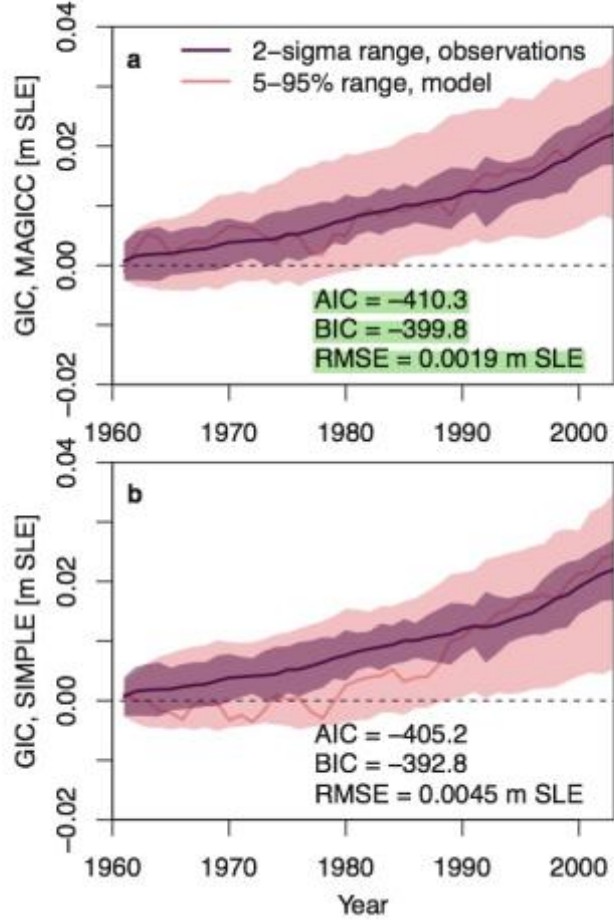

**Figure 3.** Comparison of (a) GIC-MAGICC versus (b) GIC-SIMPLE model performance in hindcasting the glaciers and ice caps (GIC) contribution to sea-level rise. GIC sea-level rise is presented relative to 1960 GIC sea level contribution. Lower values of Akaike Information Criterion (AIC), Bayesian Information Criterion (BIC), and root-mean-squared error (RMSE) indicate a better model fit to the data (Dyurgerov and Meier, 2005). These error metrics are all calculated using the maximum likelihood ensemble member, which is represented by the solid pink line. Green highlighting indicates the model structure suggested by each comparison metric.

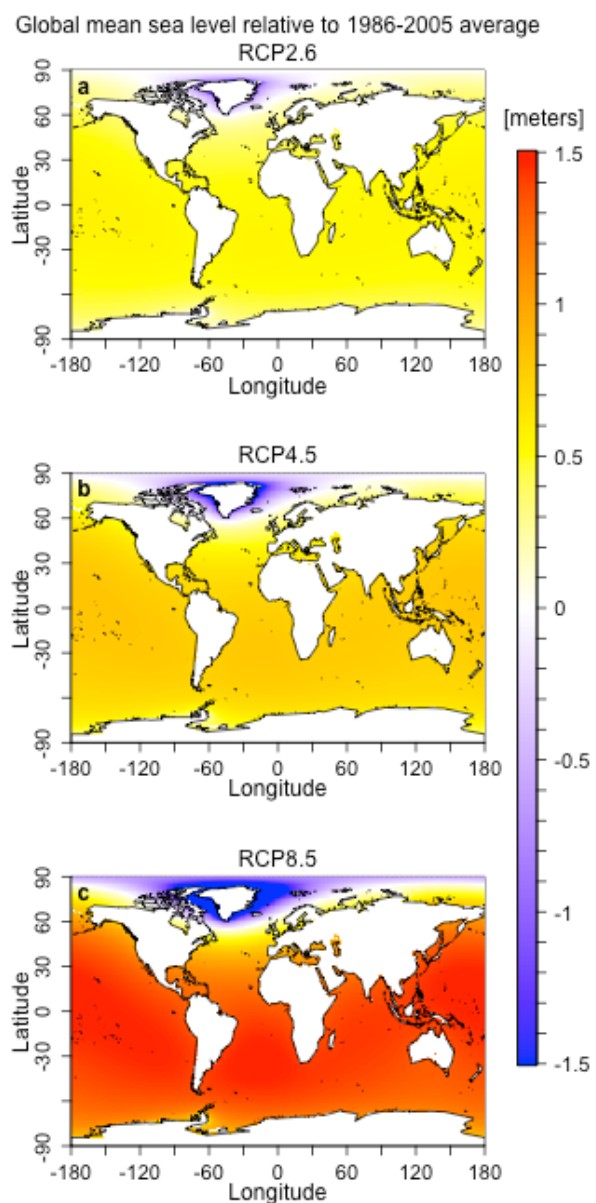

**Figure 4.** Regional projections of median sea-level changes under Representative Concentration Pathways (RCP) (a) 2.6, (b) 4.5, and (c) 8.5 in the year 2100. Sea-level rise is presented relative to 1986-2005 global mean sea level.





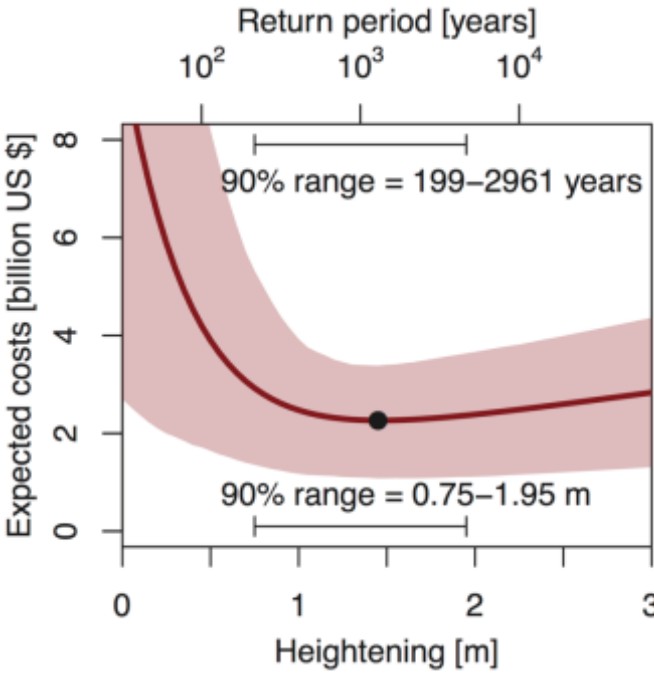

**Figure 5.** Illustrative cost-benefit analysis for the economically efficient dike heightening (lower horizontal axis) and return period (upper horizontal axis) for the north-central dike ring in New Orleans, Louisiana. The bold dot denotes the economically-efficient (i.e., cost-minimizing) solution. The shaded region gives the 90% ensemble range of trade-off curves and the bold line denotes the ensemble mean trade-off curve.