# Peer review of "BRICK v0.2, a simple, accessible, and transparent model framework for climate and regional sea-level projections"

_Geoscientific Model Development, 2016_

## Referee Comment (RC1) · Anonymous Referee #1 · 3 Mar 2017

The submitted manuscript, "BRICK, a simple, accessible, and transparent model framework for climate and regional sea-level projections," presents a good example for other geophysical modelers to follow. The authors describe a modular and transparent framework for projecting changes in regional sea level under different uncertain future scenarios, and they also give an example of how the framework can be used to plug in other modules enabling decision support based on the climate model outputs. While the flood risk management example is simplistic, it is illustrative of the potential for the BRICK model to be leveraged in a variety of useful applications. Further, the authors make a nice case for the value and importance of open-source, transparent, and simple modeling.

I believe the paper is of high quality and nearly ready for publication, but I have included a number of suggested revisions or comments on certain elements as outlined below:

P2, line 25 – difficulty can be due to a variety of things: closed platforms, reliance on databases or other inputs that are less portable than source code, etc.

P3, line 26 – Is that intended to be conservative, rather than "underconfident"? Worth explaining underconfident about what, exactly.

I am unqualified to comment on the fundamental dynamics described in section 3. However, this section provides what appears to be an appropriate level of detail, and the components are based in reputable sources and representations from other well-vetted models.

P14, 1-5 – Ability to easily recalibrate model in future with new data and/or methods is a very nice feature that should provide more longevity to the model

P18, 21-22 – "With respect to dike heightening, the expected investments are a linearly increasing function": this is not strictly accurate, as written, and should be explained more clearly. Jonkman (2009) makes a reasonable assumption that construction costs are proportional to the length of levee being constructed or upgraded, but the resulting calculations appear to show that investment costs are linear with respect to the log of the return period of level of protection provided. This is also a bit different than what readers might reasonably interpret the highlighted phrase to mean. Raw material costs when upgrading levees scale with the square of the levee height, because when raising the height, the base must also be widened.

P18, 27 – what does the "exponential flood frequency constant" represent? Is this related to the amount the probability of flooding is reduced per meter of increased dike height?

P18, 27-28 – What factors are rolled up into the net discount rate? Jonkman (2009) assumes a real interest rate, net of inflation, and then makes further reductions for

economic growth and changes in the yearly probability of flooding due to sea level rise. This is perhaps a small point because the discount rate is treated as uncertain, but if the intent is to follow Jonkman, it should be noted that the flood probability due to SLR is now endogenized in the BRICK analysis, rather than being an exogenous factor for Jonkman.

P18, 30-31 – How were the plausible ranges for each of these parameters chosen? Some of the choices seem a bit odd, such as assuming that the top end of the range for the investment cost uncertainty is 1. Why was the particular mean probability of flooding chosen? I acknowledge that this is not particularly important, given the illustrative nature of this simplified example, but some additional explanation of the experimental setup would be helpful to put it on par with the level of thoroughness given to previous sections.

4.4.2. – Given the local example, it would be nice to say something about the sea level rise encountered by Louisiana here. Otherwise, this section seems a bit out of place. In the previous section, it is stated that results (for the decision-analysis module) are presented for RCP 8.5, but then this section dives into sea level rise elsewhere in the world and also in RCPs 2.6 and 4.5. The authors may wish to consider i) removing this section, ii) making more clear that the sea level rise serves as an input into the flood risk module and integrating it better into the rest of the section 4.4 discussion, or iii) moving this section back to 4.3 or elsewhere, then mentioning the local sea level rise in Louisiana as part of 4.4.1, in relation to being an input to the flood risk module.

---

## Referee Comment (RC2) · Anonymous Referee #2 · 9 Mar 2017

The manuscript, "BRICK v0.1, a simple, accessible, and transparent model framework for climate and regional sea-level projections, describes and open-source, modular modeling framework to investigate change in global and regional sea level. The authors details the modeling framework well all with conveying the value of this type of modeling. I suggest publication with a few minor comments to be addressed below.

1. My main concern is that the model is contained in a zip file. This made it difficult to look at the structure and code without downloading the whole package. For maximum visibility and reproducibility it would be great to publish this model on github or bitbucket. This would allow for easy code review, version control, and issue tracking etc.

2. I would be useful to the reader to make it clear upfront that you are coupling multiple,

already published, models together.

3. A description of the inputs and outputs along with the spatial and temporal scales and rough run times would be useful. For example, does the model take in an emission pathway? Concentrations? Only $CO_2$?

4. Does the user have to calibrate the model? Or does the model come already calibrated?

pg1 ln17 'easier to reproduce' Easier than what?

Section 2.2.2 - are there instructions on how to incorporate new datasets?

pg7 ln14 – what data is used for this comparison?

pg7 ln14-21 – I suggest expanding this paragraph a bit more.

pg9 ln31 - are the projections in the manuscript all relative to this mean?

Figure 2 – the coloring on my end was hard to see.

———————————————————

---

## Referee Comment (RC3) · Anonymous Referee #3 · 15 Mar 2017

Manuscript Number: GMD-2016-303 Title: BRICK v0.1, a simple, accessible, and transparent model framework for climate and regional sea-level projections

General comment

This manuscript outlines design choices for the simplified contribution-based sea level model BRICK, lists the underlying equations and shows some of its features. Further, it discusses a simple application deriving regional flood risk. I expected the manuscript to be a model description paper, (which is also the chosen GMD category), but it is not. The reader is directed to another manuscript (Bakker16b, https://arxiv.org/pdf/1609.07119.pdf), which is currently under review elsewhere. Figures on calibration of sea level components as well as global sea level projections

for the RCP scenarios, which I expected in this manuscript, are instead found in Bakker16b. It is therefore difficult for me to judge what is new and original in this paper (except for the applications) and would provide sufficiently substantial advance for publication in its current form. Both the manuscripts seem to point to the same source code and I think that attribution of the code needs to be clarified. Further, on p13,L9 the authors mention that the calibration has been modified. Therefore, the reader does not have means to build trust in the calibration even when reading the Bakker16b paper. As I guess there is no possibility to merge the two manuscript (which I would find ideal), I therefore find it necessary that the title and abstract are adjusted so that it becomes clear that this is an application paper of the model (with the extra of presenting the equations, which are missing in Bakker16b). Alternatively, to make this an original contribution as model description paper, it clearly has to be highlighted what is new and different in this paper as compared to Bakker16b. I would then like to see the figures for the calibration and projections of the sea level components repeated (I expect they are not completely the same). On a positive note, I highly appreciate the effort of the authors to be as transparent as possible, providing input data, calibration data and source code. I quickly managed to reproduce the core figures. I acknowledge the open-source approach, which is missing for still too many of the climate modeling papers published. See also the specific comments.

Specific comments

Though the authors refer to Bakker16b for details on calibration, p13,L9 mentions that the calibration has been modified. This is also evident from the posterior ranges in Tables A1-A5 as compared to Bakker16b Table S3. Therefore, even with Bakker16b at hand, it is not easily possible for the reader to assess the quality of the here presented numbers. This needs thorough further discussion, see general comment above.

You shortly discuss over-parametrization but I find your argumentation not yet convincing. In p16 L8: Wouldn't a lower BIC for the full BRICK model be a stronger indicator for the full model being superior? The higher BIC than BRICK-GMSL actually hints to overparametrization, right? Also, on p16L11 you discuss that missing annual variability is of little concern. Shouldn't your more complex full BRICK model with 39 parameters better capture the dynamics and thus also better capture the shorter timescales of variability than the 13 parameter BRICK-GMSL model? Please discuss this and include some hints on where "variability got lost" and on potential improvements.

You use the model for thermal expansion that uses global mean temperature as input (equ. 15) though the DOECLIM model explicitly provides ocean heat uptake, which could be used to calculated thermal expansion. Why did you not go the DOECLIM way? Can you compare the two approaches and discuss the difference?

Your model equations 3-7 cannot be easily related to equations 8-9, which are the relevant ones for model calibration and slr contribution from Greenland. They are not in Bakker16a. Your sentence on p9L20 "SIMPLE algrebra ..." is not enough to understand the simplification from equs 3-7 to 8-9. Please outline this derivation clearly. Equ 3-7 may be moved to an Appendix together with such outline as they are not fully necessary to understand your equ 8-9 model.

Land water storage changes through dams and groundwater pumping plays a role for past and future sea level rise, see the papers of Yoshihide Wada for example. Ignoring such influences your ensemble selection as you use past global mean sea level rise as a criterion. It will also add to future sea level rise and thus flood risk. If not included in the model this should at least be discussed appropriately. It would be good to shift in ensemble members if LWS is substracted from global mean sea level rise.

Similarly, not all sea level change can be attributed to climate change since the start of industrialization as the ocean, glaciers and ice sheets all have longer memory. If I see this correctly, you assume global mean temperature change being the sole driver, thus attributing all sea level change to temperature change since preindustrial. This has been a main critique to so-called semi-empirical models and you should comment on this here or, best, do some sensitivity tests.
You do not mention how thermal expansion (or more generally: ocean dynamics) enters your regional sea level projections though it is an important contribution. I see in your code that you assume constant thermal expansion around the globe. This should a) be mentioned and b) be justified.

Equation 13, p30 is unclear to me. Sea level rise and Antarctic ice volume loss should be related by a constant factor. Instead, your right side of the equation is a sum. I think this is wrong. Please correct or explain.

One important last point: you provide the source code and the data as a zip file (though section 2.3 highlights the importance of version tracking.) Transparency and accessibility (as highlighted in sec 2.2 would profit if you'd follow your words: using one of the gitlab/github/bitbucket sites would make your code easier accessible and changes to it transparent. I think this is a precondition for publication of the manuscript if you want to keep section 2. Such repository should hold a README.md similar to your current readme, which names the additional R packages needed, i.e. Deoptim, ncdf4, gplots, fields. http://joss.theoj.org/about#reviewer_guidelines provides guidelines for such readme. A short and illustrative example with global sea level projections would be great. Why not creating a notebook for such? See https://github.com/tanyaschlusser/Jupyter-with-R/blob/master/example-Jupyter-R.ipynb as example. I guess such gitlab/github/bitbucket repository is on your plan after publication, as also indicated in Bakker16b.

Minor comments:

Section 2: Framework design As said before, I highly appreciate your efforts to be open source and transparent, but I think this section can be shortened considerably here as it does not contribute to the understanding of the model. You could address the points mentioned in a more direct way as outlined in the last paragraph of the specific comments.

There is a zoo of reference periods, including 1850-1870, 1850-1970, 1961-1990,

1986-2005 and 1960. I wonder if this could be reduced for clarity.

Timeseries figures: Think about your color scheme: pink and violet may not be the best combination.

Citations: The introduction includes a lot of references to co-authors. It is therewith a bit difficult to assess the paper's position within the field. Could you be broader?

Introduction: If you expand this to be a model description paper, I would like to see a recap of the state of the art of sea level projections. What about the past, what data is available, what can large climate models do ... ?

p1: L18: useful for uncertainty quantificiation: repeats the "pivotal role in the quantification ... of uncertainties..." of L16. Rephrase or delete this sentence L23: "aims to help mitigate": maybe two verbs would be enough. L23: "these issues": I can guess what you mean, but it is not clear. Be more precise.

p2: L9: "allotment" is this "allocation"? L32: "there is a wide range ...": I would move such outlook to the end of the paper.

p3: L10: "They simulate climate ...": they simulate global mean temperature change would be more appropriate at this level of complexity I think. L15: "drive high-risk events" suggests some physical driver. this is not true I think. rather "represent" or similar L17: "its flexibility": not clear -> "the flexibility of ..."

p4: L3: "to simulate climate change", as before: you rather try to model the response of global mean temperature to perturbations in the radiative forcing. "simulating climate change" is bigger than this. L4 rather "simulated temperature and sea level rise"

p.5: L9: " through a clear outlet for coupling to socioeconomic models": I think you talk about a stable and well documented API (application programming interface).

p7: L26: "below" can go I think L27-29: "Initial conditions ... earliest year of the simulation" These two sentences do not make sense to me. Why do you start at "certain

years" and why would you integrate backwards? If this is not the standard forward in time modeling, you should explain this in more detail.

p8: L14: Why do you not calibrate the uncertain glacier equilibrium temperature -0.15°C?

p9 L20: "SIMPLE (algebra) simplifies . . ." not clear. please rephrase and expand.

p8-p9, equations 3-7 How do these equations relate to the model you use? The relation is also not evident from Bakker16a. See specific comment above.

p10: L10: Why include the time rate of sea level change? L27: 14 parameters: I think over-parametrization should be discussed also here.

p11 L3: "Each mass ..." This is about fingerprints and valid for all contributions. I would suggest to mere it into the more general section 3.3. L13: "is the main equation ..."

p12 L18: You assume the fingerprints to be constants, they would not be so in reality. As you explain later, this assumption is ok here.

p13 L9: There seems to be a modification to the approach of Bakker16b, it is however unclear how this changes your results. See general comment.

p14 "Exchanging BRICKs and full sea-level rise module intercomparison" This heading is rather confusing to me. In the first part you talk about plugging in a global sea level model. In the second part you discuss several goodness-of-fit measures. You can be more precise in the heading. And have a subheading for the goodness of fit paragraph.

p15 L5: "this specific emulator ..." refer to Rahmstorf once again here, otherwise unclear.

p16 L8: "These mixed results ...": I think this sentence has no strong basis. You should explain better why you think your model is not overparametrized if you get "mixed results." Wouldn't a lower BIC for the full BRICK model be a stronger indicator for the full model being superior? The higher BIC actually hints towards overparametrization,

right? See also specific comments.

L11: Paragraph about variability "... missing annual variability is of little concern." You are running over this, but you should not. Your 39 parameter model captures much less short term variability than the GMSL model. You add complexity just to note that you can resolve less the dynamics of SLR? You should find a good reasoning here to justify this.

p19, paragraph 4.4.3: You should name somewhere Fig. 5 as I think that is what you are talking about here.

Fig. 2: I think you here name "BRICK-R07" what you normally call "BRICK-GMSL".

Source Code Comment:

Just to let you know how a person new to the code may address this: I had a look into the code and found the READMEs and comments within the code files, great! I did not get the model running straight away, but almost. Here is my way: First, look into ./README: Ok, I need to compile fortran files. This was easy after reading fortran/README and deleting the *so and obj/* files. I think it is better to not deliver them with the code, as they are platform dependent (at least). As I did not want to do the full calibration, I wanted to test the projections. I searched for projections and you write in ./README to have a look into /calibration/README_projections, which I did. However, the script described therein, run_BRICK.R, is not given in the repository, so I could not run the projections. I went back to the ./README, followed the text and read further about ./calibration/processingPipeline_BRICKexperiments.R, which I got running after an install.packages("ncdf4"). I adjusted the plotdir, needed to install.packages('fields') and install.packages('gplots') and could then source("analysis_and_plots_BRICKexperiments.R"). Nice!
* * *

---

## Author Comment (AC1) · 10 Apr 2017

Thank you very much for the kind words and constructive review. Below we address the open issues. We have formatted our responses in blue text to better distinguish them from the comments. The formatted PDF response file is provided as a Supplemental File, and plain text is provided here.

Yours sincerely, Tony Wong and Alexander Bakker (for the author team)

===

Comment #1

The submitted manuscript, "BRICK, a simple, accessible, and transparent model framework for climate and regional sea-level projections," presents a good example for other geophysical modelers to follow. The authors describe a modular and transparent framework for projecting changes in regional sea level under different uncertain future scenarios, and they also give an example of how the framework can be used to plug in other modules enabling decision support based on the climate model outputs. While the flood risk management example is simplistic, it is illustrative of the potential for the BRICK model to be leveraged in a variety of useful applications. Further, the authors make a nice case for the value and importance of open-source, transparent, and simple modeling.

I believe the paper is of high quality and nearly ready for publication, but I have included a number of suggested revisions or comments on certain elements as outlined below:

Reply

Thank you very much. We address the suggestions below.

===

Comment #2

P2, line 25 – difficulty can be due to a variety of things: closed platforms, reliance on databases or other inputs that are less portable than source code, etc.

Reply

We agree that it is important to stress that good coding practice is not the sole requisite for reproducibility. In response, we added the statement of:

"Studies based on simple, mechanistically-motivated models have the potential to be transparent and reproducible when presented in open platforms and when the underlying data are readily available. Yet, although there ..."

===

Comment #3

P3, line 26 – Is that intended to be conservative, rather than "underconfident"? Worth explaining underconfident about what, exactly.

Reply

Conservative estimates and underconfidence are certainly related. In our view conservative estimates are deliberately risk-adverse (e.g. by using wide uncertainty ranges) whereas underconfidence refers to the tendency to have more outcomes within the estimated probabilistic uncertainty range than expected.

We add the following short explanation: "..., e.g. by applying conservative estimates in the sense of being risk-averse"

===

Comment #4/5

I am unqualified to comment on the fundamental dynamics described in section 3. However, this section provides what appears to be an appropriate level of detail, and the components are based in reputable sources and representations from other well vetted models.

P14, 1-5 – Ability to easily recalibrate model in future with new data and/or methods is a very nice feature that should provide more longevity to the model

Reply

That is what we aimed for. Yet, we hope that the accessibility and flexibility will help others and ourselves to test alternative model choices and assumptions, as well as data.

===

Comment #6

P18, 21-22 – "With respect to dike heightening, the expected investments are a linearly increasing function": this is not strictly accurate, as written, and should be explained more clearly. Jonkman (2009) makes a reasonable assumption that construction costs are proportional to the length of levee being constructed or upgraded, but the resulting calculations appear to show that investment costs are linear with respect to the log of the return period of level of protection provided. This is also a bit different than what readers might reasonably interpret the highlighted phrase to mean. Raw material costs when upgrading levees scale with the square of the levee height, because when raising the height, the base must also be widened.

Reply

The focus of this manuscript is especially the transparency, the accessibility and flexibility of the BRICK framework. The simple approximation of Jonkman et al. (together with its extremely clear description) is designed to fit this purpose. The reviewer is, of course, absolutely correct that our description should be as clear as possible too. Besides, the description contained a small error. We rephrased it as follows: "In this simplified model, the investment costs only depend on dike heightening and are approximated by linear interpolation between data points provided by Jonkman et al. (and linear extrapolation for dike heightenings outside this range)."

===

Comment #7/8

P18, 27 – what does the "exponential flood frequency constant" represent? Is this related to the amount the probability of flooding is reduced per meter of increased dike height?

P18, 27-28 – What factors are rolled up into the net discount rate? Jonkman (2009) assumes a real interest rate, net of inflation, and then makes further reductions for economic growth and changes in the yearly probability of flooding due to sea level rise.

This is perhaps a small point because the discount rate is treated as uncertain, but if the intent is to follow Jonkman, it should be noted that the flood probability due to SLR is now endogenized in the BRICK analysis, rather than being an exogenous factor for Jonkman.

Reply

This is a great point, and in the revised manuscript we elaborate on details of the parameters of the flood risk module. Specifically, we now have added two separate paragraphs that provide a more detailed explanation about the uncertain parameters, including their assumed sampling distributions. We point to specific textual examples in our response to Comment #9, as those changes address both points.

===

Comment #9

P18, 30-31 – How were the plausible ranges for each of these parameters chosen? Some of the choices seem a bit odd, such as assuming that the top end of the range for the investment cost uncertainty is 1. Why was the particular mean probability of flooding chosen? I acknowledge that this is not particularly important, given the illustrative nature of this simplified example, but some additional explanation of the experimental setup would be helpful to put it on par with the level of thoroughness given to previous sections.

Reply

We have clarified the choices for these parameter ranges in the revised text. A summary of these motivations has been added to the revised manuscript's text, and is included below. This new text also includes a more thorough description of the flood risk parameters (addressing the reviewer's comments #7/8 above). We acknowledge that some of the ranges are somewhat ad hoc. They are meant, of course, to serve as a demonstration of model capability and not to inform on-the-ground decisions.

"The uncertain parameters considered in this cost-benefit analysis include the initial flood frequency with no heightening (y-1); the exponential flood frequency constant (m-1); the value of goods protected by the dike ring (billion US dollars); the net discount rate (%); the uncertainty in investment costs (a unitless multiplicative factor); and the land subsidence rate (m y-1). The central estimates for the exponential flood frequency constant (alpha) and the initial flood frequency with 0 heightening (p0) are taken from Van Dantzig (1956). The exponential flood frequency constant relates the increase in flood probability that results from an increase in sea level relative to the dike height. We make the assumption that this factor should scale (to first order) relatively well from Dutch case considered by Van Dantzig (1956) to the test case of New Orleans considered presently. The initial flood frequency with 0 heightening (p0) may not translate directly between these two cases, but highlights our intent for this experiment to serve as an example of future applications of the BRICK model to inform decision analyses. The admittedly ad hoc distributions assumed for alpha and p0 were selected to sample tightly around the central estimates from Jonkman et al. (2009). A more detailed treatment of this risk management problem would include using methods from extreme value theory to address the risks posed by storm surges (Coles et al. 2001).

The investment uncertainty considered in the sensitivity tests of Jonkman et al. (2009) included a base case, 50% lower, and 100% higher than the base case. We use this range for the investment uncertainty, applied as a multiplicative factor ranging from 0.5 to 2. The range for the value of good protected by the dike ring is taken from Jonkman et al (2009), where the lower bound is the lowest estimate of value of goods protected by the three dike rings considered in that work (US$5 billion), and the upper bound is the estimated combined value protected by all three dike rings (US$30 billion). The net discount rate range is centered at 4%, the estimate from Jonkman et al (2009) accounting for inflation and interest rate. Those authors' net discount rate is decreased to 2% due to economic growth (1%) and increased flooding probability due to sea-level rise (1%). Our demonstrative example endogenizes the effects of sea-level rise and accounts for parametric uncertainty in the value of good protected by the dike

ring. Hence, we center our range for the net discount rate at 4% but allow for +/-2% uncertain range. The rate of land subsidence is based on the estimates of Dixon et al. (2006), with mean 5.6 mm/y and standard deviation 2.5 mm/y. We transform this to a log-normal distribution to disallow negative rates of land subsidence.

We sample the uncertainty in these parameters via Latin hypercube, where the population size is given by the number of sea-level rise ensemble members that are present (10,671 for the control BRICK ensemble). . . ." (proceeds as in original manuscript)

Additional notes:

We also have revised the notation in Table 1 to more clearly convey how the Iunc factor translates to uncertainty in the investment costs for dike heightening. We changed the notation to Iunc in [0.5, 2], which is more precisely conveys 50% lower to 100% higher.

===

Comment #10

4.4.2. – Given the local example, it would be nice to say something about the sea level rise encountered by Louisiana here. Otherwise, this section seems a bit out of place. In the previous section, it is stated that results (for the decision-analysis module) are presented for RCP 8.5, but then this section dives into sea level rise elsewhere in the world and also in RCPs 2.6 and 4.5. The authors may wish to consider i) removing this section, ii) making more clear that the sea level rise serves as an input into the flood risk module and integrating it better into the rest of the section 4.4 discussion, or iii) moving this section back to 4.3 or elsewhere, then mentioning the local sea level rise in Louisiana as part of 4.4.1, in relation to being an input to the flood risk module.

Reply

We have revised the first sentence of Section 4.4.2 in order to make clear how the maps of regional sea level changes are related to the flood risk experiment of Section 4.4:

"In order to link projections of sea-level rise to problems of local coastal adaptation, regional sea level is projected to 2100 under the climate change scenarios of RCP2.6, 4.5, and 8.5 (Fig. 4)."

We have also revised the transition from 4.4.2 to 4.4.3 by modifying first sentence of Section 4.4.3:

"We now focus on the regional sea-level projections for the gridcell containing New Orleans, Louisiana (29 57' N, 90 4' W) under RCP8.5 (Fig. 4c), to demonstrate the use of these sea-level projections in a common local flood risk management example."

Please also note the supplement to this comment:
http://www.geosci-model-dev-discuss.net/gmd-2016-303/gmd-2016-303-AC1-
supplement.pdf

---

## Author Comment (AC2) · 10 Apr 2017

Thank you very much for the kind words and constructive review. Below we address the open issues. We have formatted our responses in blue text to better distinguish them from the comments. The formatted PDF response file is provided as a Supplemental File, and plain text is provided here.

Yours sincerely,

Tony Wong and Alexander Bakker (on behalf of the author team)

===

[Figure]

**General comment**

The manuscript, "BRICK v0.1, a simple, accessible, and transparent model framework for climate and regional sea-level projections, describes and open-source, modular modeling framework to investigate change in global and regional sea level. The authors details the modeling framework well all with conveying the value of this type of modeling. I suggest publication with a few minor comments to be addressed below.

**Reply**

Thank you very much. We address the suggestions below

===

**Comment #1**

My main concern is that the model is contained in a zip file. This made it difficult to look at the structure and code without downloading the whole package. For maximum visibility and reproducibility it would be great to publish this model on github or bitbucket. This would allow for easy code review, version control, and issue tracking etc.

**Reply**

Thank you for pointing this out – it is exactly our intent *not* to distribute the model widely using a zip file or tarball. Indeed, this would go against our stated interest in reproducibility, longevity, and transparency. Our codes are maintained on Github, and we only put a preliminary version out to accompany the GMD Discussions manuscript as a zip file. In our updated manuscript, Code and Data Availability section, we point to a Github site where the codes will be maintained for the long term:

"All BRICK v0.2 code is available at https://github.com/scrim-network/BRICK under the GNU general public open source license. Large parameter files as well as model codes forked from the repository to reproduce this work (including the sea level projections) may be found at https://download.scrim.psu.edu/Wong_etal_BRICK/."

===

Comment #2

It would be useful to the reader to make it clear upfront that you are coupling multiple, already published, models together.

Reply

You are, of course, correct. In the abstract we now have: "Here, we introduce a simple model framework (largely building on existing models) for projections of . . ."

And in the introduction: "In this model framework, we present a set of existing, well-tested, and easy-to-couple simple models for. . ."

===

Comment #3

A description of the inputs and outputs along with the spatial and temporal scales and rough run times would be useful. For example, does the model take in an emission pathway? Concentrations? Only $CO_2$?

Reply

All of the component-models are zero-dimensional, with the following exceptions. The ocean-model is a 3-layer 1D model. The Antarctic ice sheet model (DAIS) considers a two-dimensional axisymmetric geometry. These exceptions are noted in the original manuscript, Page 10 Line 7 (DAIS) and Page 7 Line 4 (ocean). In Section 2.2.4 in the original manuscript, we give rough estimates of the run times (order of thousandths of a second per 1850-present hindcast simulation).

At page 7 line 25 in the original manuscript, we point out that the sea-level rise model uses a one-year time step: "The differential equations for the GIC, GIS, AIS, and TE contributions to global mean sea level (below) are integrated in BRICK using first-order

numerical integration schemes with a one-year time step." The annual time scale can be easily adjusted.

In the revised manuscript, we have added text to make clear that the climate component uses a one-year time step as well, and state the required forcing is a radiative forcing time series: "We use a one-year time step for the DOECLIM model, and the required input to drive the model is the radiative forcing time series (W m-2)."

For projections, this uses Representative Concentration Pathways (as seen in the presentation of the results) and for the hindcasts, we use the same data as Urban and Keller (2010) and Urban et al. (2014).

The other component models are driven by global temperature and the Antarctic ice sheet contribution and the local sea-levels also require sea-level contributions from all sea-level components.

We have revised the text within Sections 3.1 and 3.2 to make these details more clear:

"DOECLIM is a zero-dimensional energy balance model coupled to a three-layer, one-dimensional diffusive ocean model."

"We adopt a simple zero-dimensional sub-model for the contribution to global sea-level rise from Glaciers and Ice Caps (GIC) from Wigley and Raper (2005)."

"BRICK uses the mechanistically-motivated, zero-dimensional SIMPLE (Simple Ice-sheet Model for Projecting Large Ensembles) model as the parameterization for the Greenland ice sheet (GIS) contribution to global mean sea level change (Bakker et al., 2016a)."

===

Comment #4

Does the user have to calibrate the model? Or does the model come already calibrated?

Reply

We will provide the larger calibrated parameter sets at a download server if users wish to use the sea-level projections showcased in the revised manuscript; we have run a larger ensemble since the initial submission, but the resulting conclusions have not changed. We hope that these projections, along with the "BRICK_LSL.R" script to fingerprint to local sea level at a user-defined latitude and longitude, will be useful for readers to incorporate numerous uncertainties in sea level projections into their own work. Thus, the model may be used already calibrated, but the model's accessibility enables easy experiments with alternative calibration schemes. Additionally, we have added a "Code Example" for using our sea-level projections and fingerprinting to yield local sea-level projections in a new README.md file, available from the Github repository.

===

Small points

pg1 ln17 'easier to reproduce' Easier than what? -> change in "easy"

Reply

Corrected in the revised Abstract.

===

Section 2.2.2 - are there instructions on how to incorporate new datasets?

Reply

We aim for a transparent easy to access framework. To test this, we need feedback from users and we will try (maybe with the help of other users) to incorporate this feedback. We have added a note about this to the "README_calibration" file in the "calibration" directory:

[Figure]

"Additional observational datasets for calibration may be introduced by making the following modifications. In the calibration directory: (1) [submodel]_readData.R – read the dataset, match the model and observational data indices (using "compute_indices" function) (2) BRICK_calib_driver.R – add to the obs.all, obs.err.all, midx.all, oidx.all, ind.norm.data lists (these tell the BRICK_assimLikelihood.R routines how to compare the model and data) (3) BRICK_assimLikelihood.R – calculate a likelihood function value for these data and add to "log.lik" routine. Note that simply adding the log-likelihood from the new dataset assumes independence between residuals from one dataset to the next. In the data directory: (1) Add the dataset, and make sure to point to this in [submodel]_readData.R"

===

pg7 ln14 – what data is used for this comparison?

pg7 ln14-21 – I suggest expanding this paragraph a bit more.

Reply

In the original manuscript, we mention these data sets at Page 7 Line 17: "We add the heteroscedastic observational error estimates from Morice et al. (2012) and Gouretski and Koltermann (2007)"

We have expanded this to specify that the data are temperature and ocean heat uptake in the revised text, as suggested: "We add the heteroscedastic observational error estimates for global mean surface temperature from Morice et al. (2012) and for ocean heat uptake from Gouretski and Koltermann (2007)"

This type of calibration approach has been used previously, and we point to those studies for further details at the end of the paragraph in question: "This type of calibration approach for DOECLIM has been implemented previously in the literature (Urban and Keller, 2010; Urban et al., 2014), and we direct the interested reader to these studies for further details."

===

pg9 ln31 - are the projections in the manuscript all relative to this mean?

Reply

Yes, that is correct.

===

Figure 2 – the coloring on my end was hard to see.

Reply

We appreciate this opportunity to improve the clarity of our figures, and have revised the color schemes for both Figure 2 and Figure 3.

Please also note the supplement to this comment:
http://www.geosci-model-dev-discuss.net/gmd-2016-303/gmd-2016-303-AC2-supplement.pdf

---

## Author Comment (AC3) · 10 Apr 2017

Thank you very much for the kind words and the critical, but constructive review. Below we address your comments. We have formatted our responses in blue text to better distinguish them from the comments. The formatted PDF response file is provided as a Supplemental File, and plain text is provided here.

Yours sincerely,

Tony Wong and Alexander Bakker (on behalf of the author team)

===

[Figure]

General comment

This manuscript outlines design choices for the simplified contribution-based sea level model BRICK, lists the underlying equations and shows some of its features. Further, it discusses a simple application deriving regional flood risk. I expected the manuscript to be a model description paper, (which is also the chosen GMD category), but it is not. The reader is directed to another manuscript (Bakker16b, https://arxiv.org/pdf/1609.07119.pdf), which is currently under review elsewhere. Figures on calibration of sea level components as well as global sea level projections for the RCP scenarios, which I expected in this manuscript, are instead found in Bakker16b. It is therefore difficult for me to judge what is new and original in this paper (except for the applications) and would provide sufficiently substantial advance for publication in its current form. Both the manuscripts seem to point to the same source code and I think that attribution of the code needs to be clarified. Further, on p13,L9 the authors mention that the calibration has been modified. Therefore, the reader does not have means to build trust in the calibration even when reading the Bakker16b paper. As I guess there is no possibility to merge the two manuscript (which I would find ideal), I therefore find it necessary that the title and abstract are adjusted so that it becomes clear that this is an application paper of the model (with the extra of presenting the equations, which are missing in Bakker16b). Alternatively, to make this an original contribution as model description paper, it clearly has to be highlighted what is new and different in this paper as compared to Bakker16b. I would then like to see the figures for the calibration and projections of the sea level components repeated (I expect they are not completely the same). On a positive note, I highly appreciate the effort of the authors to be as transparent as possible, providing input data, calibration data and source code. I quickly managed to reproduce the core figures. I acknowledge the open-source approach, which is missing for still too many of the climate modeling papers published. See also the specific comments.

Reply to general comment

Thank you for pointing out this avenue to clarify the difference in scope between this manuscript and Bakker16b. Whereas Bakker16b focuses on the model (i.e. set of equations) and its calibration, this study focuses on the code behind the model which has been specifically designed to support *transparency*, *accessibility* and *flexibility*. We note that the GMD description of a model description paper contains the statement:

"In addition to complete models, this type of paper may also describe model components and modules, as well as frameworks and utility tools used to build practical modelling systems, such as coupling frameworks or other software toolboxes with a geoscientific application." In particular, we interpret this to mean that describing and demonstrating a useful coupling framework for pre-existing models does qualify ours as a GMD model description paper.

One may argue that this should be common practice in scientific modelling and we couldn't agree more. Yet, in our assessment, this is not common practice. Transparency, accessibility and flexibility are (interrelated) modeling values that are of utmost importance for the scientific process (which obviously continues after successful peer-review and publication of a manuscript).

For example, the specific comments below contain some well justified and well considered concerns about the model choices of Bakker16b. The model values behind BRICK can facilitate discussing, exploring and testing such concerns. We hope that good coding practice, with care for the mentioned model values, will be to the advantage of the scientific modelling. As a result, this paper, in our assessment, fits nicely the category "model description paper".

Although not perfect, all reviewers express their appreciation for our attempts to be transparent, accessible and flexible. In the revised manuscript, we try to better clarify the scope, model values and coding practice. Further, we feel that the model description and the small modifications with respect should be better explained in order to

improve transparency (see replies to specific comments).

To address the reviewer's concern regarding clarity regarding the originality of the models, in the revised abstract, we now write:

"The BRICK model framework is written in R and Fortran and expands upon a recently published model setup. BRICK gives special attention to the model values of transparency, accessibility and flexibility in order to mitigate the above-mentioned issues, while . . .."

In the revised Introduction, we additionally emphasize that BRICK is built from "existing, well tested" simple models.

===

Specific comment #1

Though the authors refer to Bakker16b for details on calibration, p13,L9 mentions that the calibration has been modified. This is also evident from the posterior ranges in Tables A1-A5 as compared to Bakker16b Table S3. Therefore, even with Bakker16b at hand, it is not easily possible for the reader to assess the quality of the here presented numbers. This needs thorough further discussion, see general comment above.

Reply

We have modified the calibration relative to Bakker et al. 2016b by including a contribution from land water storage (per the reviewer's later suggestion) and using rejection sampling to join the Antarctic ice sheet model parameters (calibrated using a paleo simulation of 240,000+ years) with the rest of the model parameters (calibrated using a modern simulation from 1850-2009).

We have added this point to the overview of the sea-level rise module in the revised manuscript: "BRICK accounts for land water storage contributions to global mean sea level using mass balance trends from the International Panel on Climate Change

(IPCC) Fifth Assessment Report (AR5, Church et al., 2013) and from the work of Dieng et al. (2015)."

We have also added the land water storage term to the sea-level mass balance in Eq. (1):

dS/dt = dS_GSIC/dt + dS_GIS/dt + dS_AIS/dt + dS_TE/dt +dS_LWS/dt

where S_LWS (last term on the right-hand side) is the sea level contribution from changes in land water storage.

We have moved the sentence the reviewer mentioned (original manuscript at p13, L9) to the following paragraph, and clarify that the use of rejection sampling and subtraction of land water storage contributions (estimated from the IPCC AR5 (Church et al., 2013, Table 13.1)) are the key differences between this work and that of Bakker et al. 2016b:

"We combine these two disjoint sets of parameters to form concomitant full BRICK model parameters sets, and calibrate these to global mean sea level data (Church and White, 2011) using rejection sampling (Votaw Jr. and Rafferty, 1951). Prior to rejection sampling, contributions from land water storage are estimated using trends from the IPCC (Church et al., 2013) and subtracted from global mean sea level. When projecting global mean sea-level rise, we estimate land water storage contributions by extrapolating using the 2003-2013 trend of 0.30+/-0.18 mm/y found by Dieng et al. (2013). This approximation may not hold in reality (Wada et al., 2012), but serves as a starting point for future model developments. The use of rejection sampling and the estimation of land water storage contributions to sea level are the two aspects in which our calibration approach differs from that of Bakker et al. (2016b). In this rejection sampling step, each full BRICK parameter set is constructed by. . ."

===

Specific comment #2

You shortly discuss overparametrization but I find your argumentation not yet convincing. In p16 L8: Wouldn't a lower BIC for the full BRICK model be a stronger indicator for the full model being superior? The higher BIC than BRICK-GMSL actually hints to over parametrization, right? Also, on p16L11 you discuss that missing annual variability is of little concern. Shouldn't your more complex full BRICK model with 39 parameters better capture the dynamics and thus also better capture the shorter timescales of variability than the 13 parameter BRICK-GMSL model? Please discuss this and include some hints on where "variability got lost" and on potential improvements.

Reply

Thanks for pointing out this opportunity to clarify the exposition. You are, of course, correct, that the AIC for the full BRICK model is lower than for the Rahmstorf 2007 emulator, which indicates that the full BRICK model fits "better", but the BIC for the Rahmstorf emulator is lower, which suggests the contrary. The BIC more heavily penalizes based on the number of parameters, which we note at Page 15 Line 30 in the original manuscript. At Page 16 Line 8 of the original text, we address this mixed result, but aim to make this point clearer in the revised text by writing:

"These mixed results for the model comparison metrics indicate that using the full BRICK sea-level rise module is not unreasonably over-parameterized; if the full BRICK model were obviously over-parameterized, we would expect the AIC for the GMSL emulator experiment to be lower than for the full BRICK model."

As to the point about capturing the variability: The full BRICK model in these experiments is not directly calibrated using GMSL data. Rather, the GMSL data are invoked in the rejection sampling step that joins the paleoclimate (Antarctic ice sheet parameters) with the modern (rest of the model components' parameters) calibrations. Thus, the full BRICK model ensemble captures the individual components of sea-level rise (and temperature and ocean heat uptake), then is culled via rejection sampling to only those ensemble members which also match GMSL data. This can readily be seen in Figure 3, that the interannual variability in glaciers and ice caps contribution to sea level

is better captured. By averaging over the ensemble and the four major contributions to global mean sea level, this variability is – as expected – smoothed.

We completely agree with the reviewer that this is an important point that the original manuscript was in need of improvement. We have revised our discussion of this experiment in Section 4.2.3 of the revised manuscript:

"The full BRICK simulation does not capture the annual variation in global mean sea level that the BRICK-GMSL simulation successfully captures. This is attributed to the smoothing effect of averaging over the model ensemble the four major contributions to global mean sea level, as opposed to calibrating the BRICK-GMSL simulations directly to global mean sea level data."

===

Specific comment #3

You use the model for thermal expansion that uses global mean temperature as input (equ. 15) though the DOECLIM model explicitly provides ocean heat uptake, which could be used to calculated thermal expansion. Why did you not go the DOECLIM way? Can you compare the two approaches and discuss the difference?

Reply:

This would be a nice experiment indeed. The reason why we did not do this is the difficulty to obtain ocean heat uptake data that match the spatial and temporal resolution of the model. That means that we cannot separately test the proposed model to estimate expansion from ocean heat. In this paper, we focus on observational data sets for calibration as opposed to modeled reconstructions (which are more widely available).

===

Specific comment #4

Your model equations 3-7 cannot be easily related to equations 8-9, which are the

relevant ones for model calibration and slr contribution from Greenland. They are not in Bakker16a. Your sentence on p9L20 "SIMPLE algebra ..." is not enough to understand the simplification from equs 3-7 to 8-9. Please outline this derivation clearly. Equ 3-7 may be moved to an Appendix together with such outline as they are not fully necessary to understand your equ 8-9 model.

Reply

You are absolutely correct. Equations 3-7 cannot be easily related to equations 8-9. Our original text neglected several further approximations, hence and the two sets of equations are not fully interchangeable. We apologize for this mistake. We removed the equations 3-7 (originally intended to clarify) from the manuscript and slightly reordered the section.

===

Specific comment #5

Land water storage changes through dams and groundwater pumping plays a role for past and future sea level rise, see the papers of Yoshihide Wada for example. Ignoring such influences your ensemble selection as you use past global mean sea level rise as a criterion. It will also add to future sea level rise and thus flood risk. If not included in the model this should at least be discussed appropriately. It would be good to shift in ensemble members if LWS is subtracted from global mean sea level rise.

Reply

We thank the reviewer for this nice insight. We have revised the rejection sampling step of our calibration to global mean sea level (GMSL) data (Church and White, 2011) such that contributions from land water storage estimated from IPCC AR5 (Church et al., 2013; Table 13.1, Ch. 13, p.1151) are subtracted from the GMSL data set prior to rejection sampling.

We have added a rudimentary estimation of the land water storage contributions to

global mean sea level to our projections as well. We use the 2003-2013 trend of 0.30 +/- 0.18 mm/y from Dieng et al. (2013), and assume this trend continues to 2100. We sample annual contributions to sea level from land water storage normally with mean 0.30 mm and standard deviation 0.18 mm. This addition shifts the ensemble 5-95% range for projected GMSL by 2100 from 0.91-1.73 m to 0.95-1.74 m in RCP8.5, for example. We appreciate the suggestion and opportunity to include land water storage contributions in at least a rudimentary way in our model framework

We note the limitation of these assumptions in the revised text. Namely, that extrapolation of the trend of Dieng et al. (2013) may not hold in reality (Wada et al., 2012). The following text is added to Section 4.1:

"We combine these two disjoint sets of parameters to form concomitant full BRICK model parameters sets, and calibrate these to global mean sea level data (Church and White, 2011) using rejection sampling (Votaw Jr. and Rafferty, 1951). Prior to rejection sampling, contributions from land water storage are estimated using trends from the IPCC (Church et al., 2013) and subtracted from global mean sea level. When projecting global mean sea-level rise, we estimate land water storage contributions by extrapolating using the 2003-2013 trend of 0.30+/-0.18 mm/y found by Dieng et al. (2013). This approximation may not hold in reality (Wada et al., 2012), but serves as a starting point for future model developments. The use of rejection sampling and the estimation of land water storage contributions to sea level are the two aspects in which our calibration approach differs from that of Bakker et al. (2016b). In this rejection sampling step, each full BRICK parameter set is constructed by..."

We also point to this in the overview of the sea-level rise module in the revised manuscript in Section 3.2: "BRICK accounts for land water storage contributions to global mean sea level using mass balance trends from the International Panel on Climate Change (IPCC) Fifth Assessment Report (AR5, Church et al., 2013) and from the work of Dieng et al. (2015)."

===

Specific comment #6

Similarly, not all sea level change can be attributed to climate change since the start of industrialization as the ocean, glaciers and ice sheets all have longer memory. If I see this correctly, you assume global mean temperature change being the sole driver, thus attributing all sea level change to temperature change since preindustrial. This has been a main critique to so-called semi-empirical models and you should comment on this here or, best, do some sensitivity tests.

Reply

Only the DOECLIM model to estimate global temperature assumes that the initial (pre-industrial) temperature was close to the equilibrium temperature belonging to the then atmospheric composition. The other models are not necessarily in equilibrium at the start of the calculations.

===

Specific comment #7

You do not mention how thermal expansion (or more generally: ocean dynamics) enters your regional sea level projections though it is an important contribution. I see in your code that you assume constant thermal expansion around the globe. This should a) be mentioned and b) be justified.

Reply

Thank you for this excellent point. We are currently not aware of a method to estimate (the effect on changing) ocean dynamics (on local sea-level rise) by means of simple semi-empirical models. It may take a while before a satisfying simple model has been developed. In the meantime, emulators of GCM's may prove useful. Our stated aim with the BRICK model, however, is to avoid emulating other models but rather employ

preferentially observational data. In order to resolve these ocean dynamics, a depth- and latitudinally-resolved ocean model would be required.

We have added text to the revised manuscript at Section 3.2.4 (Thermal Expansion) to make clear these assumptions and modeling choices:

"BRICK uses a simple parameterization for the contribution of thermal expansion (TE) of the Earth's oceans to sea-level rise. We make the simplifying assumption that thermal expansion of the oceans occurs uniformly around the globe. While this is, of course, not strictly true, the next obvious step up in model complexity would be to use a vertically- and latitudinally-resolved model for thermal expansion, incorporating the DOECLIM model output for ocean heat uptake. This two-dimensional ocean model is beyond the scope of the simple model framework described presently, but an excellent subject for future work. Here, we employ a simple zero-dimensional thermal expansion emulator based on the parameterizations of the sea-level rise sub-models of (Mengel et al., 2016) and was originally used by (Grinsted et al., 2010) to model the total global mean sea level changes."

===

Specific comment #8

Equation 13, p30 is unclear to me. Sea level rise and Antarctic ice volume loss should be related by a constant factor. Instead, your right side of the equation is a sum. I think this is wrong. Please correct or explain.

Reply

We thank the reviewer for catching this typo. Indeed, they should be related by the constant factor (57 m SLE)/(V0,AIS m3). That is, the "1" in our original equation 13 should not have been there. We have corrected this error in the revised manuscript. Our apologies.

===

[Figure]

Specific comment #9

One important last point: you provide the source code and the data as a zip file (though section 2.3 highlights the importance of version tracking.) Transparency and accessibility (as highlighted in sec 2.2 would profit if you'd follow your words: using one of the gitlab/github/bitbucket sites would make your code easier accessible and changes to it transparent. I think this is a precondition for publication of the manuscript if you want to keep section 2. Such repository should hold a README.md similar to your current readme, which names the additional R packages needed, i.e. Deoptim, ncdf4, gplots, fields. http://joss.theoj.org/about#reviewer_guidelines provides guidelines for such readme. A short and illustrative example with global sea level projections would be great. Why not creating a notebook for such? See https://github.com/tanyaschlusser/Jupyter-with-R/blob/master/example-Jupyter-R.ipynb as example. I guess such gitlab/github/bitbucket repository is on your plan after publication, as also indicated in Bakker16b.

Reply

This is, again, an excellent point – it is exactly our intent *not* to distribute the model widely using a zip file or tarball. Indeed, this would go against our stated interest in reproducibility, longevity, and transparency. Our codes are maintained on Github, and we only put a preliminary version out to accompany the GMD Discussions manuscript as a zip file. In our updated manuscript Code and Data Availability section, we point to a Github site where the codes will be maintained for the long term:

"All BRICK v0.2 code is available at https://github.com/scrim-network/BRICK under the GNU general public open source license. Large parameter files as well as model codes forked from the repository to reproduce this work (including the sea level projections) may be found at https://download.scrim.psu.edu/Wong_etal_BRICK/."

We have also added a README.md file – this was a great suggestion. This file can be found in the top-layer directory at the Github link above.

===

**Minor comments**

Minor comment #1

Section 2: Framework design As said before, I highly appreciate your efforts to be open source and transparent, but I think this section can be shortened considerably here as it does not contribute to the understanding of the model. You could address the points mentioned in a more direct way as outlined in the last paragraph of the specific comments.

Reply

We appreciate the reviewer's understanding of our stated epistemic modeling values. It is specifically these sections of text, elaborating upon the needs for accessibility, transparency, efficiency and flexibility, that we feel are an important part of our message and contribution to the greater modeling community. Perhaps our release of a zip file of model codes instead of providing the Github link immediately sent the wrong message, and we have corrected this in the revised manuscript (see above reply to Specific Comment #9).

===

Minor comment #2

There is a zoo of reference periods, including 1850-1870, 1850-1970, 1961-1990,

Reply

This is true, and a result of the different observational datasets and assumed reference periods for the sub-models. Often, these sub-models include parameters whose values rely on preserving these reference periods. Our codes aim to keep track of these in a user-friendly way by passing explicitly a list object in R that keeps track of reference periods for each sub-model and dataset, and avoiding global variables (when possible)

which may hide these types of bugs.

Also – the 1850-1970 reference period was another nice typo catch, which has been corrected in the revised manuscript in Section 3.2.3 (Antarctic Ice Sheet). We apologize.

===

Minor comment #4

1986-2005 and 1960. I wonder if this could be reduced for clarity. Timeseries figures: Think about your color scheme: pink and violet may not be the best combination.

Reply

We use 1961-1990 for the hindcast reference period because all observational time series cover this period (the glaciers and ice caps data extend only to 2003 (Dyurgerov and Meier, 2005)). For the projections, we use 1986-2005 as the reference period, following the examples of Mengel et al. (2016), Church et al. (2013), and others.

We have revised the color scheme used for Figures 2 and 3.

===

Minor comment #5

Citations: The introduction includes a lot of references to co-authors. It is therewith a bit difficult to assess the paper's position within the field. Could you be broader?

Reply

Thank you for the pointer. In our view, there are two important aspects to cover in the Introduction: (1) [semi-empirical] modeling and (2) communication/connecting to decision-making. With respect to these aspects, we

(1) include references to: Hartin et al. (2015), Meinshausen et al. (2011a), Jevrejeva et al. (2016), Rahmstorf (2007), Mengel et al. (2016), and Nauels et al. (2016), as well

as the co-author references to Applegate et al. (2012), Urban et al. (2014), Urban and Keller (2010), and Bakker et al. (2016a and 2016b).

We have added a references to Grinsted et al. (2010) and Kopp et al. (2016), regarding semi-empirical modeling and uncertainty quantification.

(2) include references to: Herman et al. (2015), Weaver et al. (2013), and Lempert et al. (2004), as well as the co-author references to Hall et al. (2011), Garner et al. (2016) and Goes et al. (2011).

We have added references to Gauderis et al. (2013), Fischbach et al. (2012), and Johnson et al. (2013), regarding uncertainty and coastal risk management.

The most relevant citations in our introduction to place our model within the realm of other semi-empirical sea-level rise models are to the groundbreaking works of Mengel et al. (2016) and Nauels et al. (2016), which are not co-author citations. We note as well the need to communicate relevant references (in this case, some works our co-authors have contributed to).

===

Minor comment #6

Introduction: If you expand this to be a model description paper, I would like to see a recap of the state of the art of sea level projections. What about the past, what data is available, what can large climate models do . . . ?

Reply

This comment is addressed largely by our "Reply to general comment" above. To recap, our manuscript is well within the boundaries of a model description paper, as outlined by the GMD website:

"In addition to complete models, this type of paper may also describe model components and modules, as well as frameworks and utility tools used to build practical

modelling systems, such as coupling frameworks or other software toolboxes with a geoscientific application."

The model philosophy behind BRICK is such that it is relatively easy address this kind of questions. However, this is not the scope of this paper (see reply to general comment). The focus of the present manuscript is to present the model framework and demonstrate its flexibility and –as the reviewer's "Source Code Comment" points out– transparency and relative ease-of-use. Hence, we leave discussion of sea-level hindcasts and projections to Bakker et al. (2016b, "Sea-level projections accounting for deeply uncertain ice-sheet contributions"), which is the more appropriate manuscript to elaborate on the sea-level projections.

In our view, a comparison of a semi-empirical modeling framework such as BRICK against a large climate model (e.g., the NCAR Community Earth System Model) would be to compare apples and oranges; their purposes are quite different. We note in the Introduction the trade-off between physical model complexity and statistical model complexity (Page 2 Line 30 to Page 3 Line 3), and specify our aim to support decision-making with a nimble model capable of thoroughly exploring the low-probability, high-risk tails of distributions.

"...what data is available..." – Each sub-section of Section 3 (Model Components) includes a reference for the dataset used for calibration of BRICK. It is our intention that the assimilation of additional datasets is made simple by our transparent modeling framework.

===

Textual comments

p1: L18: useful for uncertainty quantification: repeats the "pivotal role in the quantification : : : of uncertainties: : :" of L16. Rephrase or delete this sentence L23: "aims to help mitigate": maybe two verbs would be enough. L23: "these issues": I can guess

what you mean, but it is not clear. Be more precise.

Reply

L16/18: We have rephrased L18 to read: "These qualities also make simple models useful for the characterization of risk."

L23: We have rephrased this to read "...BRICK gives special attention to the model values of transparency, accessibility, and flexibility in order to mitigate the above-mentioned issues, while..."

===

p2: L9: "allotment" is this "allocation"? L32: "there is a wide range ...": I would move such outlook to the end of the paper.

Reply

L9: We have revised the word "allotment" to read "allocation", as suggested.

L32: The aim of this paragraph is to link our epistemic modeling values to making our model useful to inform decision-making, as well as a wide range of other useful applications. In our view, this is a key aspect of the BRICK model framework (flexibility), and we would very much like to keep these key points in the Introduction.

===

p3: L10: "They simulate climate ...": they simulate global mean temperature change would be more appropriate at this level of complexity I think. L15: "drive high-risk events" suggests some physical driver. this is not true I think. rather "represent" or similar L17: "its flexibility": not clear -> "the flexibility of ..."

Reply

L10: We have revised this to read "They simulate global mean surface temperature and contributions to global mean sea-level rise."

We have also revised in the Conclusion (Sect. 5) to read "The main physics (global mean temperature and sea-level rise) codes are also..."

L15: We have replaced "drive" with "represent", as suggested. We have replaced the previous use of "represent" in this sentence with "resolve."

L17: Thank you for pointing out this inclarity. We have revised this to read "Yet, the flexibility of the BRICK model framework also enables the ..."

===

p4: L3: "to simulate climate change", as before: you rather try to model the response of global mean temperature to perturbations in the radiative forcing. "simulating climate change" is bigger than this. L4 rather "simulated temperature and sea level rise"

Reply

L3: We have revised this to read "The essence of the BRICK physical model is to simulate changes in global mean surface temperature and sea level, in response to perturbations in radiative forcing."

L4: Revised to "temperature and sea-level changes", as suggested.

===

p.5: L9: " through a clear outlet for coupling to socioeconomic models": I think you talk about a stable and well documented API (application programming interface).

Reply

In broad terms, yes, this is our intention. Within the context of the manuscript, however, we only aim to demonstrate how linking the BRICK projections for global mean sea level may be connected via the regional sea level fingerprinting to local coastal risk management problems (for example). This is a very nice suggestion and nudge into a direction to employ more sophisticated software engineering than is currently implemented in the BRICK model. Our intention is to use a high level programming language (R) as the user interface, in order to make the model accessible and comfortable to use for a broad audience.

===

p7: L26: "below" can go I think L27-29: "Initial conditions : : : earliest year of the simulation" These two sentences do not make sense to me. Why do you start at "certain years" and why would you integrate backwards? If this is not the standard forward in time modeling, you should explain this in more detail.

Reply

L26: We have removed the parenthetical comment "(below)", as suggested.

L27-29: This numerical modeling choice was motivated by the ability of this scheme to implement an initial condition for each sub-model at the reference point for the initial condition assumed by that particular sub-model. For example, as detailed in Wigley and Raper (2005), the glacier and ice cap sub-model assumes the parameter V0 is given in the year 1990. It would be possible to initialize the model in 1850, say, but this begs the question: what value should be used in this year? The most straightforward way to integrate the sub-model of Wigley and Raper (2005) is to integrate forward in time (their equation 4/5). However, the glacier data (Dyurgerov and Meier, 2005) spans 1961-2003. Solving the backwards integration problem is a trivial rearrangement of our first-order differential equations.

We understand and apologize for the ambiguity in our phrasing. To clarify this, we have revised the text here to read:

"Initial conditions are specified at a year dictated by the sub-model's assumed reference point. This differs, in general, among the sub-models and some model parameters depend on preserving this reference year. Starting from this initial condition, a first-order explicit numerical integration method integrates forward in time to the end

of the simulation and a first-order implicit (backward differentiation) method integrates backward in time to the earliest year of the simulation."

===

p8: L14: Why do you not calibrate the uncertain glacier equilibrium temperature -0.15C?

Reply

This is a good point. This was a modeling choice motivated by the need to balance computational feasibility and thoroughness. Several other temperature-related parameters exist in the Antarctic ice sheet model, and adding three more parameters (especially two to the already quite heavily parameterized AIS model) seemed to be too much.

===

p9 L20: "SIMPLE (algebra) simplifies : : :" not clear. please rephrase and expand.

Reply

See our response to "Specific comment #4", above.

===

p8-p9, equations 3-7 How do these equations relate to the model you use? The relation is also not evident from Bakker16a. See specific comment above.

Reply

See our response to "Specific comment #4", above.

===

p10: L10: Why include the time rate of sea level change? L27: 14 parameters: I think over-parametrization should be discussed also here.

Reply

L10: This is described in greater detail by Shaffer (2014) (his equations 13 and 14), but the time rate of change in sea level arises from accounting for the isotstatic adjustment of the Antarctic ice sheet, and in particular the effect of that adjustment (ice displacement) on sea level.

L27: This is a good point and overparameterization may seem to be a concern. However, our aim is to account for a wide a range of model uncertainties as possible, and constrain our simulations using observational data. Parametric uncertainty plays a large role in this accounting of uncertainty, and the Antarctic ice sheet model parameters (Shaffer, 2014; his Table 1) are no exception. If we were to assume that these parameters were known with certainty when in fact, they are not, then we would be potentially cutting off decision-relevant upper tails of the distributions of (for example) sea-level rise.

We have added a sentence to address this:

"The heavily parameterized Antarctic ice sheet module reflects our focus on including a broad range of model and observational uncertainties, and consideration of the critical role of the Antarctic ice sheet in driving substantial uncertainty in future sea levels (Church et al., 2013)."

===

p11 L3: "Each mass ..." This is about fingerprints and valid for all contributions. I would suggest to mere it into the more general section 3.3. L13: "is the main equation ..."

Reply

L3: We have revised this text to read:

"Antarctic shore-average local mean sea level functions as the input to DAIS when run as a sub-model of the coupled BRICK model. This is estimated as described in Sect.

[Figure]

3.3."

And we have added the following text to Section 3.3, as suggested:

"We couple changes in global sea level to the Antarctic ice sheet model using an Antarctic shore-average fingerprint ratio of -1.0 for the AIS contribution to global sea level, and Antarctic shore-average fingerprint factors of 1.0 for the other contributions to sea-level rise from all BRICK submodels (Slangen et al., 2014). Preliminary experiments indicated that our results are not sensitive to the precise choices of these fingerprints."

L13: Corrected to "is the main equation", as suggested.

===

p12 L18: You assume the fingerprints to be constants, they would not be so in reality. As you explain later, this assumption is ok here.

Reply

Quite true – no change necessary.

===

p13 L9: There seems to be a modification to the approach of Bakker16b, it is however unclear how this changes your results. See general comment.

Reply

Indeed this is true – see "Reply to general comment".

===

p14 "Exchanging BRICKs and full sea-level rise module intercomparison" This heading is rather confusing to me. In the first part you talk about plugging in a global sea level model. In the second part you discuss several goodness-of-fit measures. You can be more precise in the heading. And have a subheading for the goodness of fit paragraph.

Reply

This is a good observation – we agree this is unclear in the original text. In the revised text, this section heading has been updated to "Testing alternative model components: a sea-level rise module intercomparison". We have also added a subsection (4.2.2) for the goodness of fit paragraph, as suggested.

===

p15 L5: "this specific emulator ..." refer to Rahmstorf once again here, otherwise unclear.

Reply

We have revised this in the manuscript revision to read: "Note that the Rahmstorf (2007) emulator is arguably not the state-of-the-art anymore..."

===

p16 L8: "These mixed results ...": I think this sentence has no strong basis. You should explain better why you think your model is not overparametrized if you get "mixed results." Wouldn't a lower BIC for the full BRICK model be a stronger indicator for the full model being superior? The higher BIC actually hints towards overparametrization, right? See also specific comments.

Reply

See "Reply to specific comment #2" above.

===

L11: Paragraph about variability ": : : missing annual variability is of little concern." You are running over this, but you should not. Your 39 parameter model captures much less short term variability than the GMSL model. You add complexity just to note that you can resolve less the dynamics of SLR? You should find a good reasoning here to

justify this.

Reply

See "Reply to specific comment #2" above.

===

p19, paragraph 4.4.3: You should name somewhere Fig. 5 as I think that is what you are talking about here.

Reply

We thank the reviewer for pointing out this oversight. We have revised the second and third sentences of this section to read "We find the economically-efficient (i.e., cost-minimizing) dike heightening to be 1.45 m (ensemble mean; 90% range is 0.75 to 1.95 m; Fig. 5). This heightening corresponds to a return period of about 1270 years (ensemble mean; 90% range is roughly 200-3000 years; Fig. 5)."

===

Fig. 2: I think you here name "BRICK-R07" what you normally call "BRICK-GMSL".

Reply

Quite right. It has been corrected in Figure 2 of the revised manuscript.

===

Source Code Comment:

Just to let you know how a person new to the code may address this: I had a look into the code and found the READMEs and comments within the code files, great! I did not get the model running straight away, but almost. Here is my way: First, look into ./README: Ok, I need to compile fortran files. This was easy after reading fortran/README and deleting the *so and obj/* files. I think it is better to not deliver them with the code, as they are platform dependent (at least). As I did not

want to do the full calibration, I wanted to test the projections. I searched for projections and you write in ./README to have a look into /calibration/README_projections, which I did. However, the script described therein, run_BRICK.R, is not given in the repository, so I could not run the projections. I went back to the ./README, followed the text and read further about./calibration/processingPipeline_BRICKexperiments.R, which I got running after an install.packages("ncdf4"). I adjusted the plotdir, needed to install.packages('fields') and install.packages('gplots') and could then source("analysis_and_plots_BRICKexperiments.R"). Nice!

Reply

We thank the reviewer very much for the nice code review! This is precisely the level of scrutiny we hoped ours and future codes may be evaluated with.

We greatly appreciate the reviewer's comments and suggestions here. In the Github repository accompanying the revised manuscript, we have removed the *so and obj/* files (added to .gitignore), included all required routines (we apologize – this was an oversight in the codes accompanying the original manuscript), and in the top-level README file, we provide a list of the R packages needed, which may be copy-pasted into an R terminal from the README.

Please also note the supplement to this comment:
http://www.geosci-model-dev-discuss.net/gmd-2016-303/gmd-2016-303-AC3-supplement.pdf

---

## Editor Comment (EC1) · O. Marti (Editor) · 24 Apr 2017

Dear authors,

Thank you for your extensive work to respond to all reviewers remarks. A few sections of the paper have been substantially changed, and I don't feel competent to do an editor review. I'm sending back the paper to two of the reviewers.

Considering the manuscript type. I agree with you that the GMD definition of a "model description paper" spawns something wider than this title may suggest. BRICK is a "simple model framework", and your paper properly fits in this category.

Yours,

Olivier Marti GMD Topical editor

---

## Author Response (AR2)

2 June 2017

Re: GMD Paper gmd-2016-303
Authors: Tony E. Wong et al.

Dear Dr. Olivier Marti,

Thank you for your, the referees', and the entire editorial team's nice feedback and constructive suggestions. We have carefully addressed the referee's comments.

After the last iteration of the revised manuscript was submitted, we noticed a small bug in our calibration codes. This was thanks to careful review of our codes by an outside collaborator, who we have added to the acknowledgments section. We have corrected this error in the Github repository, and in the files provided. The revised analysis changes sea-level projections very little (<3 cm by 2100), and none of the main conclusions of our study have changed. We are thankful for this opportunity to improve the clarity of the manuscript, as well as place it into better context next to Bakker et al. (2016b). That manuscript is now accepted to *Nature Scientific Reports*, which we have corrected in our revised manuscript's references (now Bakker et al., 2017).

Based on the instructions provided, we have included a point-by-point response to the Referee Comments appended to this letter, as well as a tracked changes version of the revised manuscript. We have also uploaded the flattened changes file of the revised manuscript and the electronic source files to the online submission site. We have typeset the original referee comments in **black** and our responses in **blue**, to more easily distinguish the two.

We would like to take this opportunity to express our sincere thanks to the two reviewers who identified areas of our manuscript that needed corrections or modification. We would like to also thank you and the peer-review team for allowing us to resubmit a revised copy of the manuscript and for the service to our community.

We hope that the revised manuscript is found to be suitable for publication in *Geoscientific Model Development*.

Sincerely yours,

Tony E. Wong and Alexander Bakker (for the author team)
2217 Earth and Engineering Sciences Building
University Park, PA 16802
Phone: +1.216.978.8254
E-mail: twong@psu.edu

**Referee #3**

===

Still, I feel that the point of originality has to be clarified further. The authors now refer to BRICK v0.2, which only differs to Bakker16b, as stated by the authors, by the addition of a simple estimate of land water storage and rejection sampling that accounts for it. This is a very small addition to justify a new description paper. Rather, the reader should be guided to the author's intention, as I read it now, to communicate the underlying design principles better and detail the previously described model.

**Reply**
Thank you very much for the great suggestions. This is very much our intent, and we are happy for this opportunity to better communicate it.

**Action**
Addressed below.

**Comment #1**
I would therefore suggest to write in the abstract:
… Here, we describe the simple model framework BRICK (Building blocks for Relevant Ice and Climate Knowledge) v0.2 and its underlying design principles. The manuscript adds detail to an earlier published model setup and discusses the inclusion of a land water storage component. The framework largely builds on existing models and allows for projections of global mean temperature as well as regional sea levels and coastal flood risk. BRICK is written in R and Fortran. ...

and in the Introduction (L15, p2 in the tracked-changes version):
Here we describe in detail BRICK ("Building blocks for Relevant Ice and Climate Knowledge", Bakker16b) v0.2, a model framework that focuses on accessibility, transparency, and flexibility while maintaining, as much as possible, the computational efficiency that make simple models so appealing. As compared to Bakker16b, BRICK v0.2 accounts for land water storage with the other components kept unchanged ...

These changes (or similar ones) would clearly point the reader to the fact that this is not a completely new model.

**Reply**
We completely agree that these two changes better communicate our intention to describe the underlying principles in the BRICK model, as opposed to suggesting that it is completely new.

**Action**
Corrected as suggested in the revised manuscript.

===

**Comment #2**
Minor comment:
$\alpha$GIS and aGIS read very similar. Maybe exchange one of the parameter names.

**Reply**
This is a good point.

**Action**
We have exchanged aGIS for cGIS. In Table A6, we have also exchanged aGIC for cGIC (pertaining to the SIMPLE-GSIC experiment).

[revised manuscript text omitted]